# TransFool: An Adversarial Attack against Neural Machine Translation Models

## Abstract

Deep neural networks have been shown to be vulnerable to small perturbations of their inputs known as adversarial attacks. In this paper, we consider the particular task of Neural Machine Translation (NMT), where security is often critical. We investigate the vulnerability of NMT models to adversarial attacks and propose a new attack algorithm called *TransFool*. It builds on a multi-term optimization problem and a gradient projection step to compute adversarial examples that fool NMT models. By integrating the embedding representation of a language model in the proposed attack, we generate fluent adversarial examples in the source language that maintain a high level of semantic similarity with the clean samples and render the attack largely undetectable. Experimental results demonstrate that, for multiple translation tasks and different NMT architectures, our white-box attack can severely degrade the translation quality for more than 60% of the sentences while the semantic similarity between the original sentence and the adversarial example stays very high. Moreover, we show that the proposed attack is transferable to unknown target models and can fool those quite easily. Finally, based on automatic and human evaluations, our method leads to improvement in terms of success rate, semantic similarity, and fluency compared to the existing attacks both in white-box and black-box settings. Hence, TransFool permits to better characterize the vulnerability of NMT systems and outlines the necessity to design strong defense mechanisms and more robust NMT systems for real-life applications.

## 1 Introduction

The impressive performance of Deep Neural Networks (DNNs) in different areas such as computer vision (He et al., 2016) and Natural Language Processing (NLP) (Vaswani et al., 2017) has led to their widespread usage in various applications. With such an extensive usage of these models, it is important to analyze their robustness and potential vulnerabilities. In particular, it has been shown that the outputs of these models are susceptible to imperceptible changes in the input, known as adversarial attacks (Szegedy et al., 2014). Adversarial examples, which differ from the original inputs in an imperceptible manner, cause the target model to generate incorrect outputs. If these models are not robust enough to these attacks, they cannot be reliably used in applications with security requirements. To address this issue, many studies have been recently devoted to the effective generation of adversarial examples, the defense against attacks, and the analysis of the vulnerabilities of DNN models (Moosavi-Dezfooli et al., 2016; Madry et al., 2018; Ortiz-Jiménez et al., 2021).

The dominant methods to craft imperceptible attacks for continuous data, e.g., audio and image data, are based on gradient computing and various optimization strategies. However, these methods cannot be directly extended to NLP models due to the discrete nature of the tokens in the corresponding representations (i.e., words, subwords, and characters). Another challenge in dealing with textual data is the characterization of the imperceptibility of the adversarial perturbation. The $\ell_p$-norm is highly utilized in image data to measure imperceptibility but it does not apply to textual data where manipulating only one token in a sentence may significantly change the semantics. Moreover, in gradient-based methods, it is challenging to incorporate linguistic constraints in a differentiable manner. Hence, optimization-based methods are more difficult and less investigated for adversarial attacks against NLP models. Currently, most attacks in textual data are gradient-free and simply based on heuristic word replacement, which may result in *sub-optimal* performance (Alzantot et al., 2018; Ren et al., 2019; Zang et al., 2020; Jin et al., 2020; Morris et al., 2020; Guo et al., 2021; Sadrizadeh et al., 2022).

In the literature, adversarial attacks have been mainly studied for text classifiers, but less for other NLP tasks such as Neural Machine Translation (NMT) (Zhang et al., 2020b). In text classifiers, the number of output labels of the model is limited, and the adversary's goal is to mislead the target model to classify the input into any wrong class (untargeted attack) or a wrong predetermined class (targeted attack). However, in NMT systems, the output of the target model is a sequence of tokens, which is a much larger space than that of a text classifier (Cheng et al., 2020a), and it is probable that the ground-truth translation changes after perturbing the input sequence. Hence, it is important to craft meaning-preserving adversarial sentences with a low impact on the ground-truth translation.

In this paper, we propose *TransFool* to build *meaning-preserving* and *fluent* adversarial attacks against NMT models. We build a new solution to the challenges associated with gradient-based adversarial attacks against textual data. To find an adversarial sentence that is fluent and semantically similar to the input sentence but highly degrades the translation quality of the target model, we propose a multi-term optimization problem over the tokens of the adversarial example. We consider the white-box attack setting, where the adversary has access to the target model and its parameters. White-box attacks are widely studied since they reveal the vulnerabilities of the systems and are used in benchmarks. To ensure that the generated adversarial examples are imperceptibly similar to the original sentences, we incorporate a Language Model (LM) in our method in two ways. First, we consider the loss of a Causal Language Model (CLM) in our optimization problem in order to impose the syntactic correctness of the adversarial example. Second, by working with the embedding representation of LMs, instead of the NMT model, we ensure that similar tokens are close to each other in the embedding space (Tenney et al., 2019). It enables the definition of a similarity term between the respective tokens of the clean and adversarial sequences. Hence, we include a similarity constraint in the proposed optimization problem, which uses the LM embeddings. Finally, our optimization contains an adversarial term to maximize the loss of the target NMT model.

The generated adversarial example, i.e., the minimizer of the proposed optimization problem, should consist of meaningful tokens, and hence, the proposed optimization problem should be solved in a discrete space. By using a gradient projection technique, we first consider the continuous space of the embedding space and perform a gradient descent step and then, we project the resultant embedding vectors to the most similar valid token. In the projection step, we use the LM embedding representation and project the output of the gradient descent step into the nearest meaningful token in the embedding space (with maximum cosine similarity). We test our method against different NMT models with transformer structures, which are now widely used for their exceptional performance. For different NMT architectures and translation tasks, experiments show that our white-box attack can reduce the BLEU score, a widely-used metric for translation quality evaluation (Post, 2018), to half for more than 60% of the sentences while it maintains a high level of semantic similarity with the clean samples. Furthermore, we extend TransFool to black-box settings and show that it can fool unknown target models. Overall, automatic and human evaluations show that in both white-box and black-box settings, TransFool outperforms the existing heuristic strategies in terms of success rate, semantic similarity, and fluency. In summary, our contributions are as follows:

- We define a new optimization problem to compute semantic-preserving and fluent attacks against NMT models. The objective function contains several terms: adversarial loss to maximize the loss of the target NMT model; a similarity term to ensure that the adversarial example is *similar* to the original sentence; and loss of a CLM to generate *fluent* and *natural* adversarial examples.
- We propose a new strategy to incorporate linguistic constraints in our attack in a differentiable manner. Since LM embeddings provide a meaningful representation of the tokens, we use them instead of the NMT embeddings to compute the similarity between two tokens.
- We design a white-box attack algorithm, *TransFool*, against NMT models by solving the proposed optimization problem with gradient projection. Our attack, which operates at the token level, is effective against state-of-the-art transformer-based NMT models and *outperforms* prior works.
- By using the transferability of adversarial attacks to other models, we extend the proposed white-box attack to the black-box setting. Our attack is highly effective even when the *target languages* of the target NMT model and the reference model are *different*. To our knowledge, this type of transfer attack, *cross-lingual*, has not been investigated.

The rest of the paper is organized as follows. We review the related works in Section 2. In Section 3, we formulate the problem of adversarial attacks against NMT models, and propose an optimization problem to build adversarial attacks. We describe our attack algorithm in Section 4. In Section 5, we

discuss the experimental results and evaluate our algorithm against different transformer models and translation tasks. Moreover, we evaluate our attack in black-box settings and show that TransFool has very good transfer properties. Finally, the paper is concluded in Section 6.

## 2 RELATED WORK

Machine translation, an important task in NLP, is the task of automatically converting a sequence of words in a source language to a sequence of words in a target language (Bahdanau et al., 2015). By using DNN models, NMT systems are reaching exceptional performance, which has resulted in their usage in a wide variety of areas, especially in safety and security sensitive applications. But any faulty output of NMT models may result in irreparable incidents in real-world applications. Hence, we need to better understand the vulnerabilities of NMT models to perturbations of input samples, in particular to adversarial examples, to ensure *security* of applications and *robustness* of such models.

Adversarial attacks against NMT systems have been studied in recent years. First, Belinkov & Bisk (2018) show that character-level NMT models are highly vulnerable to character manipulations such as typos in a block-box setting. Similarly, Ebrahimi et al. (2018a) investigate the robustness of character-level NMT models. They propose a white-box adversarial attack based on HotFlip (Ebrahimi et al., 2018b) and greedily change the important characters to decrease the translation quality (untargeted attack) or mute/push a word in the translation (targeted attack). However, character-level manipulations can be easily detected. To circumvent this issue, many of the adversarial attacks against NMT models are rather based on word replacement. Cheng et al. (2019) propose a white-box attack where they first select random words of the input sentence and replace them with a similar word. In particular, in order to limit the search space, they find some candidates with the help of a language model and choose the token that aligns best with the gradient of the adversarial loss to cause more damage to the translation. Michel et al. (2019) and Zhang et al. (2021) find important words in the sentence and replace them with a neighbor word in the embedding space to create adversarial examples. However, these methods use heuristic strategies which may result in sub-optimal performance. There are also some other types of attacks against NMT models in the literature. In (Wallace et al., 2020), a new type of attack, i.e., universal adversarial attack, is proposed, which consists of a single snippet of text that can be added to any input sentence to mislead the NMT model. However, the added phrase is meaningless, hence easily detectable. Cheng et al. (2020a) propose Seq2Sick, a targeted white-box attack against NMT models. They introduce an optimization problem and solve it by gradient projection. The proposed optimization problem contains an adversarial loss and a group lasso term to ensure that only a few words of the sentence are modified. Although they have a projection step to the nearest embedding vector, they use the NMT embeddings, which may not preserve semantic similarity.

Other types of attacks against NMT models with different threat models and purposes have also been investigated in the literature. Some papers focus on making NMT models robust to perturbation to the inputs (Cheng et al., 2018; 2020b; Tan et al., 2021). Some other papers use adversarial attacks to enhance the NMT models in some aspects, such as word sense disambiguation (Emelin et al., 2020), robustness to subword segmentation (Park et al., 2020), and robustness of unsupervised NMT (Yu et al., 2021). In (Xu et al., 2021; Wang et al., 2021), the data poisoning attacks against NMT models are studied. Another type of attack whose purpose is to change multiple words while ensuring that the output of the NMT model remains unchanged is explored in (Chaturvedi et al., 2019; 2021). Another attack approach is presented in (Cai et al., 2021), where the adversary uses the hardware faults of systems to fool NMT models.

In summary, most of the existing adversarial attacks against NMT models are not undetectable since they are based on *character manipulation*, or they use the *NMT embedding* space to find similar tokens. Also, heuristic strategies based on *word-replacement* are likely to have sub-optimal performance. Finally, none of these attacks study the *transferability* to black-box settings. We introduce *TransFool* to craft effective and fluent adversarial sentences which are similar to the original ones.

## 3 OPTIMIZATION PROBLEM

In this section, we first present our new formulation for generating adversarial examples against NMT models, along with different terms that form our optimization problem.

**Adversarial Attack.** Consider $\mathcal{X}$ to be the source language space and $\mathcal{Y}$ to be the target language space. The NMT model $f : \mathcal{X} \rightarrow \mathcal{Y}$ generally has an encoder-decoder structure (Bahdanau et al., 2015; Vaswani et al., 2017) and aims to maximize the translation probability $p(\mathbf{y}_{\text{ref}}|\mathbf{x})$, where $\mathbf{x} \in \mathcal{X}$ is the input sentence in the source language and $\mathbf{y}_{\text{ref}} \in \mathcal{Y}$ is the ground-truth translation in the target language. To process textual data, each sentence is decomposed into a sequence of tokens. Therefore, the input sentence $\mathbf{x} = x_1 x_2 ... x_k$ is split into a sequence of $k$ tokens, where $x_i$ is a token from the vocabulary set $\mathcal{V}_{\mathcal{X}}$ of the NMT model, which contains all the tokens from the source language. For each token in the translated sentence $\mathbf{y}_{\text{ref}} = \mathbf{y}_{\text{ref},1}, ..., \mathbf{y}_{\text{ref},l}$, the NMT model generates a probability vector over the target language vocabulary set $\mathcal{V}_{\mathcal{Y}}$ by applying a softmax function to the decoder output.

The adversary is looking for an adversarial sentence $\mathbf{x}'$, which is tokenized into a sequence of $k$ tokens $\mathbf{x}' = x_1' x_2' ... x_k'$, in the source language that fools the target NMT model, i.e., the translation of the adversarial example $f(\mathbf{x}')$ is far from the true translation. However, the adversarial example $\mathbf{x}'$ and the original sentence $\mathbf{x}$ should be imperceptibly close so that the translation of the adversarial example stays similar to $\mathbf{y}_{\text{ref}}$.

As is common in the NMT models (Vaswani et al., 2017; Junczys-Dowmunt et al., 2018; Tang et al., 2020), to feed the discrete sequence of tokens into the NMT model, each token is converted to a continuous vector, known as an embedding vector, using a lookup table. In particular, let emb(.) be the embedding function that maps the input token $x_i$ to the continuous embedding vector $\text{emb}(x_i) = \mathbf{e}_i \in \mathbb{R}^m$, where $m$ is the embedding dimension of the target NMT model. Therefore, the input of the NMT model is a sequence of embedding vectors representing the tokens of the input sentence, i.e., $\mathbf{e_x} = [\mathbf{e}_1, \mathbf{e}_2, ..., \mathbf{e}_k] \in \mathbb{R}^{(k \times m)}$. In the same manner, $\mathbf{e_{x'}} = [\mathbf{e_1'}, \mathbf{e_2'}, ..., \mathbf{e_k'}] \in \mathbb{R}^{(\mathbf{k} \times \mathbf{m})}$ is defined for the adversarial example.

To generate an adversarial example for a given input sentence, we introduce an optimization problem with respect to the embedding vectors of the adversarial sentence $\mathbf{e_{x'}}$. Our optimization problem is composed of multiple terms: an adversarial loss, a similarity constraint, and the loss of a language model. An adversarial loss causes the target NMT model to generate faulty translation. Moreover, with a language model loss and a similarity constraint, we impose the generated adversarial example to be a fluent sentence and also semantically similar to the original sentence, respectively. The proposed optimization problem, which finds the adversarial example $\mathbf{x}'$ from its embedding representation $\mathbf{e_{x'}}$ by using a lookup table, is defined as follows:

$$\mathbf{x}' \leftarrow \underset{\mathbf{e}_i' \in \mathcal{E}_{\mathcal{V}_{\mathcal{X}}}}{\arg\min} \ [\mathcal{L}_{Adv} + \alpha \mathcal{L}_{Sim} + \beta \mathcal{L}_{LM}], \tag{1}$$

where $\alpha$ and $\beta$ are the hyperparameters that control the relative importance of each term. Moreover, we call the continuous space of the embedding representations the embedding space and denote it by $\mathcal{E}$, and we show the discrete subspace of the embedding space $\mathcal{E}$ containing the embedding representation of every token in the source language vocabulary set by $\mathcal{E}_{\mathcal{V}_{\mathcal{X}}}$. We now discuss the different terms of the optimization function in detail.

**Adversarial Loss.** In order to create an adversarial example whose translation is far away from the reference translation $\mathbf{y}_{\text{ref}}$, we try to maximize the training loss of the target NMT model. Since the NMT models are trained to generate the next token of the translation given the translation up until that token, we are looking for the adversarial example that maximizes the probability of wrong translation (i.e., minimizes the probability of correct translation) for the $i$-th token, given that the NMT model has produced the correct translation up to step $(i - 1)$:

$$\mathcal{L}_{Adv} = \frac{1}{l} \sum_{i=1}^{l} \log(p_f(y_{\text{ref},i}|\mathbf{e_{x'}}, \{y_{\text{ref},1}, ..., y_{\text{ref},(i-1)}\})), \tag{2}$$

where $p_f(y_{\text{ref},i}|\mathbf{e_{x'}}, \{y_{\text{ref},1}, ..., y_{\text{ref},(i-1)}\})$ is the cross entropy between the predicted token distribution by the NMT model and the delta distribution on the token $y_{\text{ref},i}$, which is one for the correct translated token, $y_{\text{ref},i}$, and zero otherwise. By minimizing $\log(p_f(.))$, normalized by the sentence length $l$, we force the output probability vector of the NMT model to differ from the delta distribution on the token $y_{\text{ref},i}$, which may cause the predicted translation to be wrong.

**Similarity Constraint.** To ensure that the generated adversarial example is similar to the original sentence, we need to add a similarity constraint to our optimization problem. It has been shown

that the embedding representation of a language model captures the semantics of the tokens (Tenney et al., 2019; Shavarani & Sarkar, 2021). Suppose that the embedding representation by a language model of the original sentence (which may differ from the NMT embedding representation $\mathbf{e_x}$) is $\mathbf{v_x} = [\mathbf{v}_1, \mathbf{v}_2, ..., \mathbf{v}_k] \in \mathbb{R}^{(k \times n)}$, where $n$ is the embedding dimension of the language model. Likewise, let $\mathbf{v_{x'}}$ denote the sequence of LM embedding vectors regarding the tokens of the adversarial example. We can define the distance between the $i$-th tokens of the original and the adversarial sentences by computing the cosine distance between their corresponding LM embedding vectors:

$$\forall i \in \{1, ..., k\}: \quad r_i = 1 - \frac{\mathbf{v}_i^\mathsf{T} \mathbf{v}_i'}{\|\mathbf{v}_i\|_2 . \|\mathbf{v}_i'\|_2}. \tag{3}$$

The cosine distance is zero if the two tokens are the same and it has larger values for two unrelated tokens. We want the adversarial sentence to differ from the original sentence in only a few tokens. Therefore, the cosine distance between most of the tokens in the original and adversarial sentence should be zero, which causes the cosine distance vector $[r_1, r_2, ..., r_k]$ to be sparse. To ensure the sparsity of the cosine distance vector, instead of the $\ell_0$ norm, which is not differentiable, we can define the similarity constraint as the $\ell_1$ norm relaxation of the cosine distance vector normalized to the length of the sentence:

$$\mathcal{L}_{Sim} = \frac{1}{k} \sum_{i=1}^{k} 1 - \frac{\mathbf{v}_i^\mathsf{T} \mathbf{v}_i'}{\|\mathbf{v}_i\|_2 . \|\mathbf{v}_i'\|_2}. \tag{4}$$

**Language Model Loss.** Causal language models are trained to maximize the probability of a token given the previous tokens. Hence, we can use the loss of a CLM, i.e., the negative log-probability, as a rough and differentiable measure for the fluency of the generated adversarial sentence. The loss of a CLM, which is normalized to the sentence length, is as follows:

$$\mathcal{L}_{LM} = -\frac{1}{k} \sum_{i=1}^{k} \log(p_g(\mathbf{v}_i' | \mathbf{v}_1', ..., \mathbf{v}_{(i-1)}')), \tag{5}$$

where g is a CLM, and $p_g(\mathbf{v}_i' | \mathbf{v}_1', ..., \mathbf{v}_{(i-1)}')$ is the cross entropy between the predicted token distribution by the language model and the delta distribution on the token $\mathbf{v}_i'$, which is one for the corresponding token in the adversarial example, $\mathbf{v}_i'$, and zero otherwise.

To generate adversarial examples against a target NMT model, we propose to solve the optimization problem (1), which contains an adversarial loss term, a similarity constraint, and a CLM loss.

## 4 TRANSFOOL ATTACK ALGORITHM

We now introduce our algorithm for generating adversarial examples against NMT models. The block diagram of our proposed attack is presented in Figure 1. We are looking for an adversarial example with tokens in the vocabulary set $\mathcal{V_X}$ and the corresponding embedding vectors in the subspace $\mathcal{E_{V_X}}$. Hence, the optimization problem (1) is discrete. The high-level idea of our algorithm is to use gradient projection to solve equation 1 in the discrete subspace $\mathcal{E_{V_X}}$.

The objective function of equation 1 is a function of NMT and LM embedding representations of the adversarial example, $\mathbf{e_{x'}}$ and $\mathbf{v_{x'}}$, respectively. Since we aim to minimize the optimization problem with respect to $\mathbf{e_{x'}}$, we need to find a transformation between the embedding space of the language model and the target NMT model. To this aim, as depicted in Figure

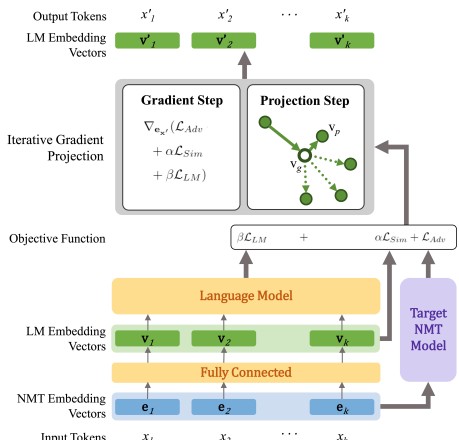

Figure 1: Block diagram of *TransFool*.

1, we propose to replace the embedding layer of a pre-trained language model with a Fully Connected (FC) layer, which gets the embedding vectors of the NMT model as its input. Then, we train the language model and the FC layer simultaneously with the causal language modeling objective. Therefore, we can compute the LM embedding vectors as a function of the NMT embedding vectors: $\mathbf{v}_i = FC(\mathbf{e}_i)$, where $FC \in \mathbb{R}^{m \times n}$ is the trained FC layer.

The pseudo-code of our attack can be found in Algorithm 1. In more detail, we first convert the discrete tokens of the sentence to continuous embedding vectors of the target NMT model, then we use the FC layer to compute the embedding representations of the tokens by the language model. Afterwards, we consider the continuous relaxation of the optimization problem, which means that we assume that the embedding vectors are in the continuous embedding space $\mathcal{E}$ instead of $\mathcal{E}_{\mathcal{V}_{\mathcal{X}}}$. In each iteration of the algorithm, we first update the sequence of embedding vectors $\mathbf{e}_{\mathbf{x}'}$ in the opposite direction of the gradient (gradient descent). Let us denote the output of the gradient descent step for the $i$-th token by $\mathbf{e}_{\mathbf{g},i}$. Then we project the resultant embedding vectors, which are not necessarily in $\mathcal{E}_{\mathcal{V}_{\mathcal{X}}}$, to the nearest token in the vocabulary set $\mathcal{V}_{\mathcal{X}}$. Since the distance in the embedding space of the LM model represents the relationship between the tokens, we use the LM embedding representations with cosine similarity metric in the projection step to find the most similar token in the vocabulary. We can apply the trained fully connected layer $FC$ to find the LM embedding representations: $\mathbf{v}_{\mathbf{g}} = FC(\mathbf{e}_{\mathbf{g}})$. Hence, the projected NMT embedding vector, $\mathbf{e}_{\mathbf{p},i}$, for the $i$-th token is:

$$\mathbf{e}_{\mathbf{p},i} = \arg\max_{\mathbf{e} \in \mathcal{E}_{\mathcal{V}_{\mathcal{X}}}} \frac{FC(\mathbf{e})^{\top} \mathbf{v}_{\mathbf{g},i}}{\|FC(\mathbf{e})\|_2 . \|\mathbf{v}_{\mathbf{g},i}\|_2}. \tag{6}$$

**Algorithm 1** TransFool Adversarial Attack

**Input:**
  $f(.)$: Target NMT model, $\mathcal{V}_{\mathcal{X}}$: Vocabulary set
  $FC$: Fully connected layer, $\mathbf{x}$: Input sentence
  $\mathbf{y}_{\mathbf{ref}}$: Ground-truth translation of $\mathbf{x}$
  $\lambda$: BLEU score ratio, $\alpha, \beta$: Hyperparameters
  $K$: Maximum No. of iterations, $\gamma$: step size
**Output:**
  $\mathbf{x}'$: Generated adversarial example
**initialization:**
  $\mathbf{s} \leftarrow$ empty set, $itr \leftarrow 0$
  $thr \leftarrow \text{BLEU}(f(\mathbf{e}_x), \mathbf{y}_{\mathbf{ref}})) \times \lambda$
  $\forall i \in \{1, ..., k\}$   $\mathbf{e}_{\mathbf{g},i}, \mathbf{e}_{\mathbf{p},i} \leftarrow \mathbf{e}_i$
**while** $itr < K$ **do**
  $itr \leftarrow itr + 1$
  **Step 1:** Gradient descent in the continuous embedding space:
  $\mathbf{e}_{\mathbf{g}} \leftarrow \mathbf{e}_{\mathbf{g}} - \gamma.\nabla_{\mathbf{e}_{\mathbf{x}'}}(\mathcal{L}_{adv} + \alpha\mathcal{L}_{Sim} + \beta\mathcal{L}_{LM})$
  $\mathbf{v}_{\mathbf{g}} \leftarrow FC(\mathbf{e}_{\mathbf{g}})$
  **Step 2:** Projection to the discrete subspace $\mathcal{E}_{\mathcal{V}_{\mathcal{X}}}$ and update if the sentence is new:
  **for** $i \in \{1, ..., k\}$ **do**
    $\mathbf{e}_{\mathbf{p},i} \leftarrow \arg\max_{\mathbf{e} \in \mathcal{E}_{\mathcal{V}_{\mathcal{X}}}} \frac{FC(\mathbf{e})^{\top} \mathbf{v}_{\mathbf{g},i}}{\|FC(\mathbf{e})\|_2 . \|\mathbf{v}_{\mathbf{g},i}\|_2}$
  **end for**
  **if** $\mathbf{e}_{\mathbf{p}}$ not in set $\mathbf{s}$ **then**
    add $\mathbf{e}_{\mathbf{p}}$ to set $\mathbf{s}$
    $\mathbf{e}_{\mathbf{g}} \leftarrow \mathbf{e}_{\mathbf{p}}$
    **if** $\text{BLEU}(f(\mathbf{e}_{\mathbf{p}}), \mathbf{y}_{\mathbf{ref}})) \leq thr$ **then**
      break (adversarial example is found)
    **end if**
  **end if**
**end while**
**return** $\mathbf{e}_{\mathbf{x}'} \leftarrow \mathbf{e}_{\mathbf{p}}$

However, due to the discrete nature of data, by applying the projection step in every iteration of the algorithm, we may face an undesirable situation where the algorithm gets stuck in a loop of previously computed steps. In order to circumvent this issue, we will only update the embedding vectors by the output of the projection step if the projected sentence has not been generated before.

We perform the gradient descent and projection steps iteratively until a maximum number of iterations is reached, or the translation quality of the adversarial example relative to the original translation quality is less than a threshold. To evaluate the translation quality, we use the BLEU score, which is a widely used metric in the literature:

$$\frac{\text{BLEU}(f(\mathbf{e}_{\mathbf{x}'}), \mathbf{y}_{\mathbf{ref}}))}{\text{BLEU}(f(\mathbf{e}_{\mathbf{x}}), \mathbf{y}_{\mathbf{ref}}))} \leq \lambda. \tag{7}$$

## 5 EXPERIMENTS

In this section, we first discuss our experimental setup, and then we evaluate TransFool against different models and translation tasks, both in white-box and black-box settings.

### 5.1 EXPERIMENTAL SETUP

We conduct experiments on the English-French (En-Fr), English-German (En-De), and English-Chinese (En-Zh) translation tasks. We use the test set of WMT14 (Bojar et al., 2014) for the En-Fr and En-De tasks, and the test set of OPUS-100 (Zhang et al., 2020a) for the En-Zh task. Some statistics of these datasets are presented in Appendix A.

We evaluate TransFool against transformer-based NMT models. To verify that our attack is effective against various model architectures, we attack the HuggingFace implementation of the Marian NMT models (Junczys-Dowmunt et al., 2018) and mBART50 multilingual NMT model (Tang et al., 2020).

As explained in Section 4, the similarity constraint and the LM loss of the proposed optimization problem require an FC layer and a CLM. To this aim, for each NMT model, we train an FC layer and a CLM (with GPT-2 structure (Radford et al., 2019)) on WikiText-103 dataset. We note that the input of the FC layer is the target NMT embedding representation of the input sentence.

Table 1: Performance of white-box attack against different NMT models.

| Task | Method | Marian NMT | | | | | | mBART50 | | | | | |
|---|---|---|---|---|---|---|---|---|---|---|---|---|---|
| | | ASR↑ | RDBLEU↑ | RDchrF↑ | Sim.↑ | Perp.↓ | TER↓ | ASR↑ | RDBLEU↑ | RDchrF↑ | Sim.↑ | Perp.↓ | TER↓ |
| En-Fr | TransFool | **69.38** | **0.57** | **0.23** | **0.85** | 182.45 | 13.91 | **60.68** | **0.53** | **0.22** | 0.84 | **121.12** | **10.58** |
| | kNN | 36.53 | 0.36 | 0.16 | 0.82 | 389.78 | 19.15 | 30.84 | 0.29 | 0.11 | **0.85** | 336.47 | 21.03 |
| | Seq2Sick | 27.01 | 0.21 | 0.16 | 0.75 | **175.31** | **13.97** | 25.53 | 0.19 | 0.13 | 0.75 | 151.92 | 13.55 |
| En-De | TransFool | **69.49** | **0.65** | **0.23** | **0.84** | 165.53 | 13.57 | **62.87** | **0.61** | **0.22** | 0.83 | **134.90** | **11.07** |
| | kNN | 39.22 | 0.40 | 0.17 | 0.82 | 441.62 | 19.42 | 35.99 | 0.39 | 0.12 | **0.86** | 375.32 | 21.22 |
| | Seq2Sick | 35.60 | 0.31 | 0.21 | 0.67 | 290.32 | 18.13 | 35.59 | 0.31 | 0.20 | 0.66 | 265.62 | 18.18 |
| En-Zh | TransFool | **73.82** | **0.74** | **0.31** | **0.88** | 102.49 | 11.82 | **57.50** | **0.67** | **0.26** | 0.90 | **74.75** | **7.77** |
| | kNN | 31.12 | 0.33 | 0.18 | 0.86 | 180.27 | 15.95 | 27.25 | 0.32 | 0.14 | 0.90 | 160.27 | 16.58 |
| | Seq2Sick | 28.76 | 0.26 | 0.25 | 0.73 | 161.84 | 17.48 | 24.25 | 0.31 | 0.18 | 0.78 | 105.42 | 13.58 |

To find the minimizer of our optimization problem (1), we use the Adam optimizer (Kingma & Ba, 2014) with step size $\gamma = 0.016$. Moreover, we set the maximum number of iterations to 500. Our algorithm has three parameters: coefficients $\alpha$ and $\beta$ in the optimization function (1), and the relative BLEU score ratio $\lambda$ in the stopping criteria (7). We set $\lambda = 0.4$, $\beta = 1.8$, and $\alpha = 20$. We chose these parameters experimentally according to the ablation study, which is available in Appendix B, in order to optimize the performance in terms of success rate, semantic similarity, and fluency.

We compare our attack with (Michel et al., 2019), which is a white-box untargeted attack against NMT models.[1] We only consider one of their attacks, called *kNN*, which substitutes some words with their neighbors in the embedding space; the other attack considers swapping the characters, which is too easy to detect. We also adapted *Seq2Sick* (Cheng et al., 2020a), a targeted attack against NMT models based on an optimization problem in the NMT embedding space, to our untargeted setting.

For evaluation, we report different performance metrics: **(1) Attack Success Rate (ASR)**, which measures the rate of successful adversarial examples. Similar to (Ebrahimi et al., 2018a), we define the adversarial example as successful if the BLEU score of its translation is *less than half* of the BLEU score of the original translation. **(2) Relative decrease of translation quality**, by measuring the translation quality in terms of *BLEU score*[2] and *chrF* (Popović, 2015). We denote these two metrics by **RDBLEU** and **RDchrF**, respectively. We choose to compute the *relative decrease* in translation quality so that scores are comparable across different models and datasets (Michel et al., 2019). **(3) Semantic Similarity (Sim.)**, which is computed between the original and adversarial sentences and commonly approximated by the *universal sentence encoder* (Yang et al., 2020)[3]. **(4) Perplexity score (Perp.)**, which is a measure of the fluency of the adversarial example computed with the perplexity score of *GPT-2 (large)*. **(5) Token Error Rate (TER)**, which measures the imperceptibility by computing the rate of tokens modified by an adversarial attack.

## 5.2 RESULTS OF THE WHITE-BOX ATTACK

Now we evaluate TransFool in comparison to kNN and Seq2Sick against different NMT models. Table 1 shows the results in terms of different evaluation metrics.[4] Overall, our attack is able to decrease the BLEU score of the target model to less than half of the BLEU score of the original translation for more than 60% of the sentences for all tasks and models (except for the En-Zh mBART50 model, where ASR is 57.50%). Also, in all cases, semantic similarity is more than 0.83, which shows that our attack can maintain a high level of semantic similarity with the clean sentences.

In comparison to the baselines, TransFool obtains a higher success rate against different model structures and translation tasks, and it is able to reduce the translation quality more severely. Since the algorithm uses the gradients of the proposed optimization problem and is not based on token replacement, TransFool can highly degrade the translation quality. Furthermore, the perplexity score of the adversarial example generated by TransFool is much less than the ones of both baselines (except for the En-Fr Marian model, where it is a little higher than Seq2Sick), which is due to the

---

[1]Code of (Cheng et al., 2019; 2020b), untargeted white-box attacks against NMTs, is not publicly available.

[2]We use case-sensitive SacreBLEU (Post, 2018) on detokenized sentences.

[3]We use the multilingual version since we are dealing with multiple languages.

[4]We discard the sentences whose original BLEU score is zero to prevent improving the results artificially. We should also note that all results are computed after the re-tokenization of the adversarial example. Since we are generating the adversarial example at the token-level, there is a small chance that, when the generated adversarial example is converted to text, the re-tokenization does not produce the same set of tokens.

Table 2: Adversarial examples* against mBART50 (En-De) generated by different methods.

| Sentence | Text |
|---|---|
| Org. | In Oregon, planners **are** experimenting with giving drivers different choices. |
| Ref. Trans. | In Oregon experimentieren die Planer damit, Autofahrern eine Reihe von Auswahlmöglichkeiten zu geben. |
| Org. Trans. | In Oregon experimentieren Planer damit, Fahrern verschiedene Wahlen zu geben. |
| Adv. TransFool | In Oregon, planners **were** experimenting with giving drivers different choices. |
| Trans. | In Oregon experimentierten Planer mit der Bereitstellung unterschiedlicher Wahlmöglichkeiten für Fahrer. |
| Adv. kNN | **in** Oregon, planners **nemmeno** experimenting with**kjer** driver**.** different choices**,** |
| Trans. | in Oregon. Planer nemmeno experimenteren mitkjer Fahrer. verschiedene Wahlen, |
| Adv. Seq2Sick | In **acontece**, planners are **studying** with **Kivakapis against decisions,** |
| Trans. | In acontece studieren Planer mit Kivakapis gegen Entscheidungen, |

*Adversarial perturbed tokens are in red, and the perturbations by TransFool are in blue in the original sentence. The changes in the translation that are the direct results of the perturbation are in brown, while the changes that are due to the failure of the target model are in orange.

integration of the LM embeddings and the LM loss term in the optimization problem. Moreover, the token error rate of our attack is lower than both baselines, and the semantic similarity is preserved better by TransFool in almost all cases since we use the LM embeddings instead of the NMT ones for the similarity constraint. While kNN can also maintain semantic similarity, Seq2Sick does not perform well in this criterion. We also computed similarity by BERTScore (Zhang et al., 2019) and BLEURT-20 (Sellam et al., 2020) that highly correlate with human judgments in Appendix D, which shows that TransFool is better than both baselines in maintaining the semantics. Moreover, as presented in Appendix D.2, the *successful* attacks by the baselines, as opposed to TransFool, are not semantic-preserving or fluent sentences. Finally, the complete setup and results of our human evaluation are presented in Appendix H, which also shows the superiority of TransFool.

We also compare the runtime of TransFool and that of the two baselines. In each iteration of our proposed attack, we need to perform a back-propagation through the target NMT model and the language model to compute the gradients. Also, in some iterations (27 iterations per sentence on average), a forward pass is required to compute the output of the target NMT model to check the stopping criteria. For the Marian NMT (En-Fr) model, on a system equipped with an NVIDIA A100 GPU, it takes 26.45 seconds to generate adversarial examples by TransFool. On the same system, kNN needs 1.45 seconds, and Seq2Sick needs 38.85 seconds to generate adversarial examples for less effective adversarial attacks, however.

Table 2 shows some adversarial examples against mBART50 (En-De). In comparison to the baselines, TransFool makes smaller changes to the sentence. The generated adversarial example is a correct English sentence, and it is similar to the original sentence. However, kNN and Seq2Sick generate adversarial sentences that are not necessarily natural or similar to the original sentences. More examples generated by TransFool, kNN, and Seq2Sick can be found in Appendix D.2. We also provide some adversarial sentences when we do not use the LM embeddings in our algorithm in order to show the importance of this component.

Indeed, TransFool outperforms both baselines in terms of success rate. It is able to generate more natural adversarial examples with a lower number of perturbations (TER) and higher semantic similarity with the clean samples in almost all cases. A complete study of hyperparameters and the effect of using LM embeddings instead of NMT embeddings for computing similarity on TransFool performance is presented in Appendix B and C, respectively.

## 5.3 PERFORMANCE IN BLACK-BOX ATTACK SETTINGS

In practice, the adversary's access to the learning system may be limited. Hence, we propose to analyze the performance of TransFool in a black-box scenario. It has been shown that adversarial attacks often transfer to another model that has a different architecture and is even trained with different datasets (Szegedy et al., 2014). By utilizing this property of adversarial attacks, we extend TransFool to the black-box scenario. We consider that we have complete access to one NMT model (the reference model), including its gradients. We implement the proposed gradient-based attack in algorithm 1 with this model. However, for the stopping criteria of the algorithm, we query the black-box target NMT model to compute the BLEU score. We can also implement the black-box transfer attack in the case where the source languages of the reference model and the target model are the same, but their target languages are different. Since Marian NMT is faster and lighter than mBART50, we use it as the reference model and evaluate the performance of the black-box attack against mBART50. We compare the performance of TransFool with WSLS (Zhang et al., 2021), a

Table 4: Performance of black-box attack, when the target language is different.

| Task | Marian NMT | | | | | | mBART50 | | | | | |
|---|---|---|---|---|---|---|---|---|---|---|---|---|
| | ASR↑ | RDBLEU↑ | RDchrF↑ | Sim.↑ | Perp.↓ | #Queries↓ | ASR↑ | RDBLEU↑ | RDchrF↑ | Sim.↑ | Perp.↓ | #Queries↓ |
| En-De → En-Fr | 60.53 | 0.55 | 0.22 | 0.84 | 169.49 | 24 | 61.68 | 0.56 | 0.22 | 0.84 | 169.51 | 23 |
| En-Fr → En-De | 66.22 | 0.63 | 0.22 | 0.84 | 198.04 | 23 | 63.86 | 0.63 | 0.21 | 0.84 | 195.50 | 24 |

black-box untargeted attack against NMT models based on word-replacement (the choice of back-translation model used in WSLS is investigated in Appendix F). We also evaluate the performance of kNN and Seq2Sick in the black-box settings by attacking mBART50 with the adversarial example generated against Marian NMT (in the white-box settings). The results are reported in Table 3. We also report the performance when attacking Google Translate, some generated adversarial samples, and similarity performance computed by BERTScore and BLEURT-20 in Appendix E.

In all tasks, with a few queries to the target model, our black-box attack achieves better performance than the white-box attack against the target model (mBART50) but a little worse performance than the white-box attack against the reference model (Marian NMT). In all cases, the success rate, token error rate, and perplexity of TransFool are better than all baselines (except for the En-Fr task, where perplexity is a little higher than Seq2Sick). The

Table 3: Performance of black-box attack against mBART50.

| Task | Method | ASR↑ | RDBLEU↑ | RDchrF↑ | Sim.↑ | Perp.↓ | TER↓ | #Queries↓ |
|---|---|---|---|---|---|---|---|---|
| En-Fr | TransFool | **70.19** | **0.58** | 0.22 | **0.85** | 175.39 | **17.08** | 27 |
| | kNN | 33.74 | 0.33 | 0.15 | 0.82 | 383.71 | 22.57 | - |
| | Seq2Sick | 25.97 | 0.21 | 0.14 | 0.75 | **173.63** | 21.13 | - |
| | WSLS | 56.21 | **0.58** | **0.27** | 0.84 | 214.23 | 31.30 | 1423 |
| En-De | TransFool | **66.76** | **0.65** | **0.22** | 0.84 | **167.54** | **16.73** | 23 |
| | kNN | 36.70 | 0.39 | 0.16 | 0.82 | 435.02 | 22.34 | - |
| | Seq2Sick | 32.17 | 0.29 | 0.20 | 0.67 | 286.67 | 26.59 | - |
| | WSLS | 44.33 | 0.50 | 0.19 | **0.86** | 219.32 | 29.12 | 1262 |
| En-Zh | TransFool | **63.27** | 0.71 | 0.27 | **0.88** | **100.14** | **14.76** | 36 |
| | kNN | 26.89 | 0.31 | 0.17 | 0.86 | 176.34 | 17.07 | - |
| | Seq2Sick | 23.65 | 0.30 | 0.23 | 0.73 | 162.67 | 25.17 | - |
| | WSLS | 40.00 | **0.72** | **0.52** | 0.83 | 186.44 | 32.35 | 1782 |

ability of TransFool and WSLS to maintain semantic similarity is comparable and better than both other baselines. However, WSLS has the highest token error rate, which makes the attack detectable. The effect of TransFool on BLEU score is larger than that of the other methods, and its effect on chrF metric comes after WSLS (except for the En-DE task, where RDchrF of TransFool is the best).

Regarding the complexity, TransFool requires only a few queries to the target model for translation, while WSLS queries the model more than a thousand times, which is costly and may not be feasible in practice. For the En-Fr task, on a system equipped with an NVIDIA A100 GPU, it takes 43.36 and 1904.98 seconds to generate adversarial examples by TransFool and WSLS, respectively, which shows that WSLS is very time-consuming.

We also analyze the transferability of the generated adversarial examples to a black-box NMT model with the same source language but a different target language. Since we need a dataset with the same set of sentences for different language pairs, we use the validation set of WMT14 for En-Fr and En-De tasks. Table 4 shows the results for two cases: Marian NMT or mMBART50 as the target model. We use Marian NMT as the reference model with a different target language than that of the target model. In all settings, the generated adversarial examples are highly transferable to another NMT model with a different target language (i.e., they have high attack success rate and large semantic similarity). The high transferability of TransFool shows that it is able to capture the common failure modes in different NMT models, which can be dangerous in real-world applications.

## 6 CONCLUSION

In this paper, we proposed *TransFool*, a white-box adversarial attack against NMT models, by introducing a new optimization problem solved by an iterative method based on gradient projection. We utilized the embedding representation of a language model to impose a similarity constraint on the adversarial examples. Moreover, by considering the loss of a language model in our optimization problem, the generated adversarial examples are more fluent. Extensive automatic and human evaluations show that TransFool is highly effective in different translation tasks and against different NMT models. Our attack is also transferable to black-box settings with different structures and even different target languages. In both white-box and black-box scenarios, TransFool obtains improvement over the baselines in terms of success rate, semantic similarity, and fluency. It is important to analyze adversarial attacks against NMT models such as TransFool to find the vulnerabilities of NMT models, measure their robustness, and eventually build more robust NMT models.

**Ethics Statement**    We introduced TransFool, an adversarial attack against NMT models, with the motivation of revealing the vulnerabilities of NMT models and paving the way for designing stronger defenses and building robust NMT models in real-life scenarios. While it remains a possibility that a threat actor may misuse our attack, we do not condone using our method with the intent of attacking a real NMT system.

**Reproducibility Statement**    The source code will be publicly available as soon as possible to help reproduce our results. Moreover, Appendix G contains the license information and more details of the assets (datasets, codes, and models).

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

# Supplementary Material
## TransFool: An Adversarial Attack against Neural Machine Translation Models

### ABSTRACT

In this supplementary material, we first provide some statistics of the evaluation datasets in Section A. The ablation study of the hyperparameters of TransFool is presented in Section B. We investigate the effect of the LM embedding representation on TransFool and kNN in Section C. More results of the white-box attack are reported in D: the results of other similarity metrics (Section D.1), performance over successful attacks (Section D.2), and some generated adversarial examples (Section D.4). Section E provides more experiments on the black-box attack: the performance of attacking *Google Translate* (Section E.1), results of other similarity metrics (Section E.2), and some generated adversarial examples (Section E.3). We discuss the effect of the back-translation model choice on WSLS in Section F. Finally, the license information and more details of the assets (datasets, codes, and models) are provided in Section G.

## A  SOME STATISTICS OF THE DATASETS

Some statistics, including the number of samples, the Average length of the sentences, and the translation quality of Marian NMT and mBART50, of the evaluation datasets, i.e., OPUS100 (En-Zh) WMT14 (En-FR) and (En-De), are reported in table 5.

Table 5: Some statistics of the evaluation datasets.

| Dataset | Average Length | #Test Samples | Marian NMT | | mBART50 | |
|---|---|---|---|---|---|---|
| | | | BLEU | chrF | BLEU | chrF |
| En-Fr WMT14 | 27 | 3003 | 39.88 | 64.94 | 36.17 | 62.66 |
| En-De WMT14 | 26 | 3003 | 27.72 | 58.50 | 25.66 | 57.02 |
| En-Zh OPUS-100 | 18 | 2000 | 33.11 | 50.98 | 29.27 | 41.92 |

## B  ABLATION STUDY

In this Section, we analyze the effect of different hyperparameters (including the coefficients $\alpha$ and $\beta$ in our optimization problem (1), the step size of the gradient descent $\gamma$, and the relative BLEU score ratio $\lambda$ in the stopping criteria Eq. (7)) on the white-box attack performance in terms of success rate, semantic similarity, and perplexity score.

In all the experiments, we consider English to French Marian NMT model and evaluate over the first 1000 sentences of the test set of WMT14. The default values for the hyperparameters are as follows, except for the hyperparameter that varies in the different experiments, respectively: $\alpha = 20$, $\beta = 1.8$, $\gamma = 0.016$, and $\lambda = 0.4$.

**Effect of the similarity coefficient $\alpha$.**  This hyperparameter determines the strength of the similarity term in the optimization problem (1). Figure 2a shows the effect of $\alpha$ on the performance of our attack. By increasing the similarity coefficient of the proposed optimization problem, we are forcing our algorithm to find adversarial sentences that are more similar to the original sentence. Therefore, as shown in Figure 2a, larger values of $\alpha$ result in higher semantic similarity. However, in this case, it is harder to fool the NMT model, i.e., lower attack success rate, RDBLEU, and RDchrF. Moreover, it seems that, since the generated adversarial examples are more similar to the original sentence, they are more natural, and their perplexity score is lower.

**Effect of the language model loss coefficient $\beta$.**  We analyze the impact of the hyperparameter $\beta$, which controls the importance of the language model loss term in the proposed optimization

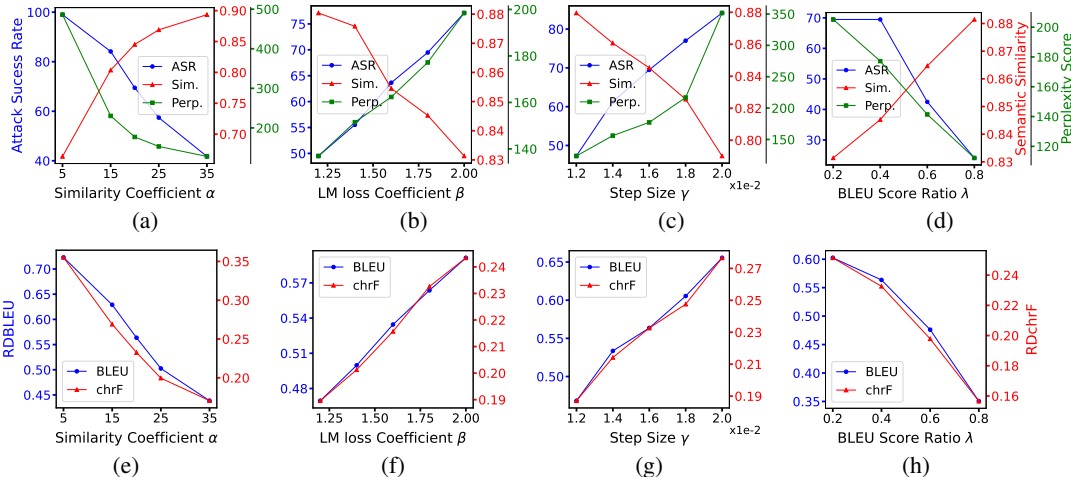

Figure 2: Effect of different hyperparameters on the performance of TransFool.

problem, in Figure 2b. By increasing this coefficient, we weaken the effect of the similarity term, i.e., the generated adversarial examples are less similar to the original sentence. As a result, the success rate and the effect on translation quality, i.e., RDBLEU and RDchrF, increase.

**Effect of the step size $\gamma$.** The step size of the gradient descent step of the algorithm can impact the performance of our attack, which is investigated in Figure 2c. Increasing the step size results in larger movement in the embedding space in each iteration of the algorithm. Hence, the generated adversarial examples are more aggressive, which results in lower semantic similarity and higher perplexity scores. However, we can find adversarial examples more easily and achieve a higher attack success rate, RDBLEU, and RDchRF.

**Effect of the BLEU score ratio $\lambda$.** This hyperparameter determines the stopping criteria of our iterative algorithm. Figure 2d studies the effects of this hyperparameter on the performance of our attack. As this figure shows, a higher BLEU score ratio causes the algorithm to end in earlier iterations. Therefore, the changes applied to the sentence are less aggressive, and hence, we achieve higher semantic similarity and a lower perplexity score. However, the attack success rate, RDBLEU, and RDchrF decrease since we make fewer changes to the sentences.

## C    EFFECT OF THE LM EMBEDDING REPRESENTATION

Table 6 shows the results of Trans-Fool and kNN when we use LM embeddings or NMT embeddings for measuring similarity between two tokens.[5] The LM embeddings result in lower perplexity and higher semantic similarity for both methods, which demonstrates the importance of this component in generating meaning-preserving fluent adversarial examples.

Table 6: Performance of white-box attack against Marian NMT (En-Fr) with/without language model embeddings.

| Method | ASR↑ | RDBLEU↑ | RDchrF↑ | Sim.↑ | Perp.↓ |
|---|---|---|---|---|---|
| TransFool w/ LM Emb. | 69.48 | 0.56 | 0.23 | 0.85 | 177.20 |
| TransFool w/ NMT Emb. | 68.27 | 0.57 | 0.26 | 0.78 | 193.32 |
| kNN w/ LM Emb. | 32.13 | 0.32 | 0.15 | 0.85 | 246.52 |
| kNN w/ NMT Emb. | 36.65 | 0.35 | 0.16 | 0.82 | 375.84 |

---

[5]In order to have a fair comparison, we fine-tuned hyperparameters of Transfool, in the case when we do not use LM embeddings, to have a similar attack success rate.

# D    MORE RESULTS ON THE WHITE-BOX ATTACK

## D.1    SEMANTIC SIMILARITY COMPUTED BY OTHER METRICS

To better assess the ability of adversarial attacks in maintaining semantic similarity, we can compute the similarity between the original and adversarial sentences using other metrics such as BERTScore (Zhang et al., 2019) and BLEURT-20 (Sellam et al., 2020). It is shown in (Zhang et al., 2019) that BERTScore correlates well with human judgments. BLEURT-20 is also shown to correlates better with human judgment than traditional measures (Freitag et al., 2021). The results are reported in Table 7. These results indicate that the TransFool is indeed more capable of preserving the semantics of the input sentence. In the two cases where kNN has better similarity by using the Universal Sentence Encoder (USE) (Yang et al., 2020), the performance of TransFool is better in terms of BERTScore and BLEURT-20.

Table 7: Similarity performance of white-box attacks.

| Task | Method | Marian NMT | | | mBART50 | | |
|------|--------|------|-----------|-------------|------|-----------|-------------|
| | | USE↑ | BERTScore↑ | BLEURT-20 ↑ | USE↑ | BERTScore↑ | BLEURT-20 ↑ |
| En-Fr | TransFool | **0.85** | **0.95** | **0.65** | 0.84 | **0.96** | **0.70** |
| | kNN | 0.82 | 0.94 | 0.61 | **0.85** | 0.93 | 0.67 |
| | Seq2Sick | 0.75 | 0.94 | 0.60 | 0.75 | 0.94 | 0.66 |
| En-De | TransFool | **0.84** | **0.96** | **0.67** | 0.83 | **0.95** | **0.69** |
| | kNN | 0.82 | 0.94 | 0.61 | **0.86** | 0.93 | 0.67 |
| | Seq2Sick | 0.67 | 0.93 | 0.52 | 0.66 | 0.92 | 0.58 |
| En-Zh | TransFool | **0.88** | **0.96** | **0.67** | **0.90** | **0.97** | **0.76** |
| | kNN | 0.86 | 0.95 | 0.66 | **0.90** | 0.95 | 0.72 |
| | Seq2Sick | 0.73 | 0.94 | 0.54 | 0.78 | 0.95 | 0.67 |

## D.2    PERFORMANCE OVER SUCCESSFUL ATTACKS

The evaluation metrics of the successful adversarial examples that strongly affect the translation quality are also important, and they show the capability of the adversarial attack. Hence, we evaluate TransFool, kNN, and Seq2Sick only over the successful adversarial examples.[6] The results for the white-box setting are presented in Table 8. By comparing this Table and Table 1, which shows the results on the whole dataset, we can see that TransFool performance is *consistent* among successful and unsuccessful attacks. Moreover, successful adversarial examples generated by TransFool are still semantically similar to the original sentences, and their perplexity score is low. However, the successful adversarial examples generated by Seq2Sick and kNN do not preserve the semantic similarity and are not fluent sentences; hence, they are *not valid* adversarial sentences.

Table 8: Performance of white-box attack over successful adversarial examples.

| Task | Method | Marian NMT | | | | | | mBART50 | | | | | |
|------|--------|------|---------|--------|-------|-------|------|------|---------|--------|-------|-------|------|
| | | ASR↑ | RDBLEU↑ | RDchrF↑ | Sim.↑ | Perp.↓ | TER↓ | ASR↑ | RDBLEU↑ | RDchrF↑ | Sim.↑ | Perp.↓ | TER↓ |
| En-Fr | TransFool | 69.38 | 0.66 | 0.26 | 0.83 | 229.75 | 15.33 | 60.68 | 0.66 | 0.27 | 0.82 | 164.52 | 12.56 |
| | kNN | 36.53 | 0.70 | 0.30 | 0.76 | 746.89 | 24.52 | 30.84 | 0.72 | 0.28 | 0.77 | 691.64 | 28.05 |
| | Seq2Sick | 27.01 | 0.72 | 0.40 | 0.56 | 648.92 | 25.28 | 25.53 | 0.74 | 0.41 | 0.53 | 556.61 | 25.16 |
| En-De | TransFool | 69.49 | 0.72 | 0.25 | 0.83 | 191.51 | 14.54 | 62.87 | 0.73 | 0.26 | 0.81 | 169.76 | 12.66 |
| | kNN | 39.22 | 0.75 | 0.29 | 0.77 | 675.01 | 23.07 | 35.99 | 0.75 | 0.23 | 0.81 | 574.68 | 25.75 |
| | Seq2Sick | 35.60 | 0.78 | 0.40 | 0.53 | 659.90 | 25.67 | 35.59 | 0.78 | 0.40 | 0.52 | 612.22 | 26.67 |
| En-Zh | TransFool | 73.82 | 0.76 | 0.34 | 0.87 | 112.28 | 12.83 | 57.50 | 0.73 | 0.31 | 0.88 | 99.08 | 9.86 |
| | kNN | 31.12 | 0.72 | 0.29 | 0.80 | 355.25 | 22.55 | 27.25 | 0.76 | 0.27 | 0.85 | 295.53 | 23.58 |
| | Seq2Sick | 28.76 | 0.72 | 0.46 | 0.58 | 437.49 | 26.84 | 24.25 | 0.79 | 0.44 | 0.60 | 292.55 | 25.59 |

## D.3    TRADE-OFF BETWEEN SUCCESS RATE AND SIMILARITY/FLUENCY

The results in our ablation study B show that there is a trade-off between the quality of adversarial example, in terms of semantic-preservation and fluency, and the attack success rate. As studied in

---

[6]As defined in Section 5, the adversarial example is successful if the BLEU score of its translation is less than half of the BLEU score of the original translation.

(Morris et al., 2020), we can filter adversarial examples with low quality based on hard constraints on semantic similarity and the number of added grammatical errors caused by adversarial perturbations.

We can analyze the trade-off between success rate and similarity/fluency by setting different thresholds for filtering adversarial examples. If we evaluate the similarity by the sentence encoder suggested in (Morris et al., 2020), the success rate with different threshold values for similarity in the case of Marian (En-Fr) is depicted in Figure 3b. By considering only the adversarial examples with a similarity higher than a threshold, the success rate decreases as the threshold increases, and the quality of the adversarial examples increases. Similarly, we can do the same analysis for fluency. As suggested in (Morris et al., 2020), we count the grammatical errors by LanguageTool (Naber et al., 2003) for the original sen-

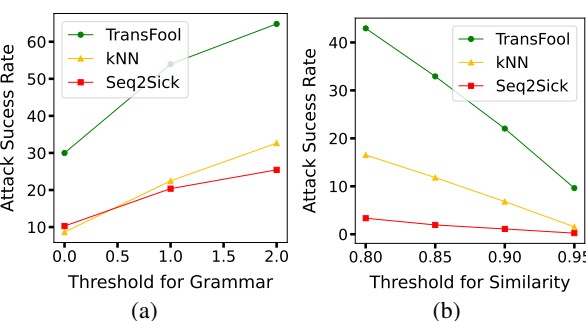

Figure 3: Tradeoff between success rate and Similarity/fluency. The left figure shows the effect of acceptable number of added grammar errors by adversarial perturbation. The right figure shows the effect of similarity threshold.

tences and the adversarial examples. Figure 3a depicts the success rate for different thresholds of the number of added grammatical errors caused by adversarial perturbations.

These analyses show that with tighter constraints, we can generate better adversarial examples while the success rate decreases. All in all, according to these results, TransFool outperforms the baselines for different thresholds of similarity and grammatical errors.

## D.4    MORE ADVERSARIAL EXAMPLES

In this Section, we present more adversarial examples generated by TransFool, kNN, and Seq2Sick. In order to show the effect of using LM embeddings on the performance of TransFool, we also include the generated adversarial examples against English to French Marian NMT model when we do not use LM embeddings. In all these tables, the tokens modified by TransFool are written in **blue** in the original sentence, and the modified tokens by different adversarial attacks are written in **red** in their corresponding adversarial sentences. Moreover, the changes made by the adversarial attack to the translation that are not directly related to the modified tokens are written in orange, while the changes that are the direct result of modified tokens are written in brown.

As can be seen in the examples presented in Tables 9 and 10, TransFool makes smaller changes to the sentence. The generated adversarial example is a correct English sentence, and it is similar to the original sentence. However, kNN, Seq2Sick, and our method with the NMT embeddings make changes that are perceptible, and the adversarial sentences are not necessarily similar to the original sentence. The higher semantic similarity of the adversarial sentences generated by TransFool is due to the integration of LM embeddings and the LM loss in the proposed optimization problem. We should highlight that TransFool is able to make changes to the adversarial sentence translation that are not directly related to the modifications of the original sentence but are the result of the NMT model failure.

Other examples against different tasks and models are presented in Tables 11 to 16.

Table 9: Adversarial examples against Marian NMT (En-Fr) by various methods (white-box).

| Sentence | BLEU | Text |
|---|---|---|
| Org. | | The most eager is **Oregon**, which is enlisting 5,000 drivers in the country's biggest experiment. |
| Ref. Trans. | | Le plus déterminé est l'Oregon, qui a mobilisé 5 000 conducteurs pour mener l'expérience la plus importante du pays. |
| Org. Trans. | 21.66 | Le plus avide est l'Oregon, qui recrute 5 000 pilotes dans la plus grande expérience du pays. |
| Adv. TransFool | | The most eager is**Quebec**, which is enlisting 5,000 drivers in the country's biggest experiment. |
| Trans. | 7.71 | Le Québec, qui fait partie de la plus grande expérience du pays, compte 5 000 pilotes. *(some parts are not translated at all.)* |
| Adv. w/ NMT Emb. | | The most eager is**Custom**, which is enlisting **Disk** drivers in the country's **editions Licensee**. |
| Trans. | 6.54 | Le plus avide estCustom, qui recrute des pilotes de disque dans les éditions du pays Licencié. |
| Adv. kNN | | **Theve** eager is Oregon, **C aren** enlisting 5,000 drivers in **theau**'s biggest experiment. |
| Trans. | 5.93 | Theve avide est Oregon, C sont enrôlés 5 000 pilotes dans la plus grande expérience de Theau. |
| Adv. Seq2Sick | | The most **buzz** is **FREE**, which is **chooseing Games comments** in the country's great **developer**. |
| Trans. | 10.31 | Le plus buzz est GRATUIT, qui est de choisir Jeux commentaires dans le grand développeur du pays. |

Table 10: Adversarial examples against Marian NMT (En-Fr) by various methods (white-box).

| Sentence | BLEU | Text |
|---|---|---|
| Org. | | "**They** are in the process of abandoning and killing off emergency units that **were** reformed less than five years ago," he believes. |
| Ref. Trans. | | "Ils sont en train de vider et d'asphyxier des urgences qui ont été rénovées il y a moins de cinq ans", estime-t-il. |
| Org. Trans. | 37.53 | « Ils sont en train d'abandonner et de tuer des unités d'urgence qui ont été réformées il y a moins de cinq ans », croit-il. |
| Adv. TransFool | | "**People** are in the process of abandoning and killing off emergency units that **been** reformed less than five years ago," he believes. |
| Trans. | 23.83 | « Les gens abandonnent et tuent les unités d'urgence réformées il y a moins de cinq ans », croit-il. *(some parts are not translated.)* |
| Adv. w/ NMT Emb. | | "**Manager** are in the process of abandoning and killing off emergency units that were **celebrating** less than five years ago," he believes. |
| Trans. | 27.66 | « Le gestionnaire est en train d'abandonner et de tuer des unités d'urgence qui célébraient il y a moins de cinq ans », croit-il. |
| Adv. kNN | | "They are in the process of abandoning and killing off emergency **allotment** that were reformed**voir8)** five years ago," **States** believes. |
| Trans. | 21.20 | « Ils sont en train d'abandonner et de tuer les allocations d'urgence qui ont été réformées il y a cinq ans8 », estime-t-il. |
| Adv. Seq2Sick | | "They are in the process of abandoning and **shot off** emergency units that were **CSIS** less than five years ago," he believes. |
| Trans. | 33.58 | « Ils sont en train d'abandonner et de tuer des unités d'urgence qui étaient le SCRS il y a moins de cinq ans », croit-il. |

Table 11: Adversarial examples against Marian NMT (En-Fr) by various methods (white-box).

| Sentence | BLEU | Text |
|---|---|---|
| Org. | | Discovering neighbourhoods, our architecture**,** our environment are reference points, |
| Ref. Trans. | | La découverte de quartiers, notre architecture, nos lieux sont des repères. |
| Org. Trans. | 29.25 | Découvrir les quartiers, notre architecture, notre environnement sont des points de référence, |
| Adv. TransFool | | Discovering neighbourhoods, our architecture **Whose** our environment are reference points, |
| Trans. | 7.96 | Découverte des quartiers, de notre architecture dont notre environnement est des points de référence, |
| Adv. kNN | | Discovering neighbourhoods, **these infrastructure**, **we** environment **is** reference points, |
| Trans. | 6.61 | Découverte des quartiers, de ces infrastructures, l'environnement est un point de référence, |
| Adv. Seq2Sick | | **awayingevaluations**, our architecture, our **energy** are reference points, |
| Trans. | 20.33 | l'élimination des évaluations, notre architecture, notre énergie sont des points de référence, |

Table 12: Adversarial examples against Marian NMT (En-De) by various methods (white-box).

| Sentence | BLEU | Text |
|---|---|---|
| Org. | | The **devices**, which track every mile a motorist drives and transmit that information to bureaucrats, are at the center of a **controversial** attempt in Washington and **state** planning offices to overhaul the **outdated** system for funding America's major roads. |
| Ref. Trans. | | Die Geräte, die jeden gefahrenen Kilometer aufzeichnen und die Informationen an die Behörden melden, sind Kernpunkt eines kontroversen Versuchs von Washington und den Planungsbüros der Bundesstaaten, das veraltete System zur Finanzierung US-amerikanischer Straßen zu überarbeiten. |
| Org. Trans. | 23.65 | Die Geräte, die jede Meile ein Autofahrer fährt und diese Informationen an Bürokraten weiterleitet, stehen im Zentrum eines umstrittenen Versuchs in Washington und in den staatlichen Planungsbüros, das veraltete System zur Finanzierung der großen Straßen Amerikas zu überarbeiten. |
| Adv. TransFool | | The **vehicles**, which track every mile a motorist drives and transmit that information to bureaucrats, are at the center of a **unjustified** attempt in Washington and **city** planning offices to overhaul the **clearer** system for funding America's major roads. |
| Trans. | 9.36 | Die *Fahrzeuge*, die jede Meile ein Autofahrer fährt und diese Informationen an Bürokraten weiterleitet, stehen im Zentrum eines *ungerechtfertigten* Versuchs in Washington und in den *Stadtplanungsbüros*, das *klarere* System zur Finanzierung der *amerikanischen Hauptstraßen* zu überarbeiten. |
| Adv. kNN | | The devices **in** which track every mile a motorist drives and transmit that **M** to bureaucrats, are **07:0** the center of a controversial attempt in Washington and state planning offices to overhaul the outdated **Estate** for funding America's major roads. |
| Trans. | 7.79 | Die *Vorrichtungen*, *in denen* jede Meile ein Autofahrer fährt und diese *M* an Bürokraten *überträgt*, sind *07:0* das Zentrum eines umstrittenen Versuchs in Washington und staatlichen Planungsbüros, das veraltete *Estate für* die Finanzierung der *amerikanischen Hauptstraßen* zu überarbeiten. |
| Adv. Seq2Sick | | The devices, which **road** every**ably** a motorist drives and transmit that information to **walnut socialisms**, are at the center of a **Senate** attempt in Washington and state planning offices to**establishment** the outdated system for funding America's major **paths**. |
| Trans. | 22.48 | Die Geräte, die *allgegenwärtig* ein Autofahrer *antreibt* und diese Informationen an *Walnusssozialismen überträgt*, stehen im Zentrum eines *Senatsversuchs* in Washington und in den staatlichen Planungsbüros, das veraltete System zur Finanzierung der *wichtigsten Wege* Amerikas *einzurichten*. |

Table 13: Adversarial examples against Marian NMT (En-Zh) by various methods (white-box).

| Sentence | BLEU | Text |
|---|---|---|
| Org. | | And what **your** husband said... **if** Columbus had done it, we'd all be Indians. |
| Ref. Trans. | | 你丈夫说的... 要是哥伦布没发现美洲,我们现在就都是印第安人了 |
| Org. Trans. | 25.58 | 你丈夫说的话... 如果哥伦布做到了我们都会是印第安人 |
| Adv. TransFool | | And **With** your husband said... if Columbus had done it, we'd all be Indians. |
| Trans. | 0.0 | 你丈夫说如果哥伦布做到了我们都会是印第安人 *(some parts are not translated.)* |
| Adv. kNN | | And what your husband said... if Columbus had**60,** we' **Nineteen** all **it** Indians. |
| Trans. | 24.45 | 你丈夫说的话... 如果哥伦布有60" 我们19个印度人 |
| Adv. Seq2Sick | | And **completing** your **penalties** said... if **timely** had done it, we'd all be **briefed**. |
| Trans. | 22.09 | 完成你的处罚说... 如果及时完成,我们都会得到简报 |

Table 14: Adversarial examples against mBART50 (En-Fr) crafted by various methods (white-box).

| Sentence | BLEU | Text |
|---|---|---|
| Org. | | Wearing a wingsuit, he flew past over the famous Monserrate Sanctuary at 160km/h. The sanctuary is located at an altitude of over 3000 meters and numerous spectators had gathered there to watch his exploit. |
| Ref. Trans. | | Equipé d'un wingsuit, il est passé à 160 km/h au-dessus du célèbre sanctuaire Monserrate, situé à plus de 3 000 mètres d'altitude, où de nombreux badauds s'étaient rassemblés pour observer son exploit. |
| Org. Trans. | 27.33 | Il a survolé à 160 km/h le célèbre sanctuaire de Monserrate, situé à une altitude de plus de 3000 mètres, où de nombreux spectateurs se sont réunis pour assister à son exploit. |
| Adv. TransFool | | Wearing a wingsuit, he flew past over the famous **Interesserrage** Sanctuary at 160km/h. The sanctuary is located at an altitude of over 3000 meters and numerous spectators had gathered there to watch his exploit. |
| Trans. | 6.16 | Le sanctuaire est situé à une altitude de plus de 3000 mètres *et* de nombreux spectateurs se sont réunis pour assister à son exploit. *(first part of the sentence is not translated at all.)* |
| Adv. kNN | | Wearing a wingsuit**.** he flew past over the famous Monserrate Sanctuary at 160km/h. The sanctuary is located at **anzu opinionstitude** of over **8000** meters and numerous spectators had gathered there **the** watch his exploit. |
| Trans. | 23.80 | Il a survolé le célèbre sanctuaire de Monserrate à 160 km/h. Le sanctuaire est situé à une altitude de plus de *8000* mètres *et* de nombreux spectateurs se sont *rassemblés là* pour *observer* son exploit. |
| Adv. Seq2Sick | | Wearing a wingsuit, he flew past over the famous Monserrate Sanctuary at 160km/h. The sanctuary is located at an altitude of **over74** meters and numerous spectators had gathered there to watch his exploit. |
| Trans. | 32.52 | Il a survolé à 160 km/h le célèbre sanctuaire de Monserrate, situé à plus de *74* mètres d'altitude, où de nombreux spectateurs se sont réunis pour assister à son exploit. |

Table 15: Adversarial examples against mBART50 (En-De) crafted by various methods (white-box).

| Sentence | BLEU | Text |
|---|---|---|
| Org. | | In Oregon, planners **are** experimenting with giving drivers different choices. |
| Ref. Trans. | | In Oregon experimentieren die Planer damit, Autofahrern eine Reihe von Auswahlmöglichkeiten zu geben. |
| Org. Trans. | 22.38 | In Oregon experimentieren Planer damit, Fahrern verschiedene Wahlen zu geben. |
| Adv. TransFool | | In Oregon, planners **were** experimenting with giving drivers different choices. |
| Trans. | 6.27 | In Oregon experimentierten Planer mit der Bereitstellung unterschiedlicher Wahlmöglichkeiten für Fahrer. |
| Adv. kNN | | **in** Oregon, planners **nemmeno** experimenting with**kjer** driver**.** different choices**,** |
| Trans. | 3.94 | in Oregon, Planer nemmeno experimenteren mitkjer Fahrer. verschiedene Wahlen, |
| Adv. Seq2Sick | | In **acontece**, planners are **studying** with **Kivakapis against decisions,** |
| Trans. | 3.22 | In acontece studieren Planer mit Kivakapis gegen Entscheidungen, |

Table 16: Adversarial examples against mBART50 (En-Zh) crafted by various methods (white-box).

| Sentence | BLEU | Text |
|---|---|---|
| Org. | | Delegations are requested to submit the names of their representatives to the Secretary of the Preparatory Committee, **Ms.** Vivian Pliner-Josephs (room **S-29**5**0**E; fax: (21**2**) 96**3-5**935). |
| Ref. Trans. | | 请各代表团将其代表姓名送交给筹备委员会秘书VivianPliner-Josephs女士(S-2950E室;电传:(212)963-5935)。 |
| Org. Trans. | 61.63 | 请各代表团向筹备委员会秘书VivianPliner-Josephs(S-2950E室;传真:(212)963-5935)提出代表的姓名。 |
| Adv. TransFool | | Delegations are requested to submit the names of their representatives to the Secretary of the Preparatory Committee, **Mr.** Vivian Pliner-Josephs (room **C-29**3**0**E; fax: (21**1**) 96 **25-3**0935). |
| Trans. | 9.55 | 请各代表团将其代表的姓名提交筹备委员会秘书维维安·普林纳-约瑟夫斯先生(房间C-2930E;传真:(211)9625-30935)。 |
| Adv. kNN | | Delegations are requested to submit the names of their representatives **that** the Secretary of the Preparatory Committee, Ms. Vivian**Pliner-Joseph,** (room S-2950 •**,** fax: (212) 963-5935). |
| Trans. | 54.37 | 请各代表团向筹备委员会秘书VivianPliner-Joseph(S-2950室;传真:(212)963-5935)递交代表的姓名。 |
| Adv. Seq2Sick | | Delegations are requested to submit the names of their representatives to the Secretary of the Preparatory Committee, Ms.**jadan** Pliner-Josephs (room S-2950E; **599**: 212 96 **2010**,935. |
| Trans. | 13.40 | 请各代表团将其代表的姓名提交筹备委员会秘书贾丹·普林纳-约塞夫斯女士(S-2950E室;599:2129620010,935)。 |

# E  MORE RESULTS ON THE BLACK-BOX ATTACK

## E.1  ATTACKING GOOGLE TRANSLATE

To evaluate the effect of different attacks in practice, we attack Google Translate[7] by TransFool, kNN, and Seq2Sick. Since querying Google Translate is limited per day, we were not able to attack with WSLS, which requires high number of queries. Table 17 presents the performance of the English to French translation task. The results demonstrate that adversarial sentences crafted by Trans-Fool can degrade the translation quality more while preserving the semantics better. The perplexity score and word error rate of TransFool compete with those metrics of Seq2Sick, but Seq2Sick is not meaning-preserving and is less effective.

Table 17: Performance of black-box attack against Google Translate (En-Fr).

| Method | ASR↑ | RDBLEU↑ | RDchrF↑ | Sim.↑ | Perp.↓ | WER↓ |
|---|---|---|---|---|---|---|
| TransFool | **67.83** | **0.55** | **0.23** | **0.85** | 184.35 | 20.85 |
| kNN | 37.22 | 0.35 | 0.17 | 0.82 | 389.45 | 30.24 |
| Seq2Sick | 23.49 | 0.20 | 0.15 | 0.75 | **174.88** | **20.34** |

Table 18: Performance of TransFool black-box attack against Google Translate (En-De), when the target language is different..

| Task | ASR↑ | RDBLEU↑ | RDchrF↑ | Sim.↑ | Perp.↓ | WER↓ |
|---|---|---|---|---|---|---|
| En-Fr → En-De | 67.42 | 0.65 | 0.26 | 0.85 | 198.56 | 20.78 |

We also performed the cross-lingual black-box attack. We consider Marian NMT (En-Fr) as the reference model and attack En-De Google Translate. The results for TransFool are reported in Table 18.

## E.2  SEMANTIC SIMILARITY COMPUTED BY OTHER METRICS

Similar to the white-box attack, we compute the similarity between the adversarial and original sentences by BERTScore and BLEURT-20, since they correlate well with human judgments. The similarity performance of Trans-Fool and WSLS[8] in the black-box settings are demonstrated in Table 19. According to Table 19, TransFool is better at maintaining semantic similarity. It may be because we used LM embeddings instead of the NMT ones in the similarity constraint.

Table 19: Similarity performance of black-box attacks.

| Task | Method | USE↑ | BERTScore↑ | BLEURT-20↑ |
|---|---|---|---|---|
| En-Fr | TransFool | **0.85** | **0.95** | **0.66** |
| | WSLS | 0.84 | 0.93 | 0.58 |
| En-De | TransFool | 0.84 | **0.96** | **0.67** |
| | WSLS | **0.86** | 0.94 | 0.61 |
| En-Zh | TransFool | **0.88** | **0.96** | **0.68** |
| | WSLS | 0.83 | 0.93 | 0.56 |

## E.3  SOME ADVERSARIAL EXAMPLES

We also present some adversarial examples generated by TransFool and WSLS, in the black-box setting, in Tables 20 to 22. In these tables, the tokens modified by TransFool are written in **blue** in the original sentence, and the modified tokens by different adversarial attacks are written in **red** in their corresponding adversarial sentences. Moreover, the changes made by the adversarial attack to the translation that are not directly related to the modified tokens are written in orange, while the changes that are the direct result of modified tokens are written in brown.

These examples show that modifications made by TransFool are less detectable, i.e., the generated adversarial examples are more natural and similar to the original sentence. Moreover, TransFool makes changes to the translation that are not the direct result of the modified tokens of the adversarial sentence.

---

[7]We should note that since we do not have a tokenizer, we compute Word Error Rate (WER) instead of Token Error Rate (TER).

[8]The results of kNN and Seq2Sick are not reported since they are transfer attacks, and their performance is already reported in Table 7.

Table 20: Adversarial examples against mBART50 (En-Fr) crafted by various methods (black-box).

| Sentence | BLEU | Text |
|---|---|---|
| Org. | | It is therefore not surprising that he should be holding a mask in the promotional **photography** for L'Invitation **au** Voyage, by Louis Vuitton, of which he is the new face. |
| Ref. Trans. | | Rien d'étonnant à ce qu'il ait donc un masque à la main dans la photographie de la campagne L'Invitation au Voyage, de la marque Louis Vuitton, dont il incarne le nouveau visage. |
| Org. Trans. | 31.15 | Il n'est donc pas surprenant qu'il tienne une masque dans la photographie promotionnelle de L'Invitation au Voyage, de Louis Vuitton, dont il est le nouveau visage. |
| Adv. TransFool | | It is therefore not surprising that he should be holding a mask in the promotional **painting** for L'Invitation **sans** Voyage, by Louis Vuitton, of which he is the new face. |
| Trans. | 13.22 | Il n'est donc pas surprenant qu'il tienne une masque dans la peinture promotionnelle de Louis Vuitton pour L'Invitation sans Voyage, dont il est le nouveau visage. |
| Adv. WSLS | | **me** is therefore not surprising that he should be holding a **wig inthe** the **promos portraiture** for L'Invitation au Voyage, by **gary** Vuitton, of which he is the **fresh** face. |
| Trans. | 21.21 | Je ne suis donc pas surpris qu'il porte une paruche dans le portrait promotionnel de l'Invitation au Voyage, de Gary Vuitton, dont il est le nouveau visage. |
| Adv. kNN | | It is therefore not surprising that he should be holding a mask in the promotional photography for L'In**processedation** au Voyage, by Louis **gooduitton**, of which he is **associating** new face**Connection** |
| Trans. | 13.82 | Il n'est donc pas surprenant qu'il tienne une masque dans la photographie promotionnelle de L'In processe-dation au Voyage, de Louis gooduitton, dont il associe un nouveau visageConnection |
| Adv. Seq2Sick | | It is therefore not **pasture** that he should be holding a **hidden** in the **unclean goodness** for L'Invitation au Voyage, by Louis Vuitton, of which he is the new face. |
| Trans. | 28.75 | Il ne s'agit donc pas d'un pâturage qu'il doit tenir caché dans la bonté insalubre pour L'Invitation au Voyage, de Louis Vuitton, dont il est le nouveau visage. |

Table 21: Adversarial examples against mBART50 (En-De) crafted by various methods (black-box).

| Sentence | BLEU | Text |
|---|---|---|
| Org. | | This really is **a** must for our nation. |
| Ref. Trans. | | Das ist wirklich ein Muss für unser Land. |
| Org. Trans. | 61.05 | Das ist wirklich ein Muss für unsere Nation. |
| Adv. TransFool | | This really is **his** must for our nation. |
| Trans. | 20.16 | Das ist wirklich seine Pflicht für unsere Nation. |
| Adv. WSLS | | This really **becomes** a must **outfor** our nation. |
| Trans. | 33.03 | Das wird wirklich ein Muss für unsere Nation. |
| Adv. kNN | | This **realities and** a **requisiteAstr** our nation. |
| Trans. | 4.77 | Diese Realitäten und eine notwendige Astrologie unserer Nation. |
| Adv. Seq2Sick | | This really is a must for our **imperfect**. |
| Trans. | 61.05 | Das ist wirklich ein Muss für unsere Unvollkommenheit. |

Table 22: Adversarial examples against mBART50 (En-Zh) crafted by various methods (black-box).

| Sentence | BLEU | Text |
|---|---|---|
| Org. | | (c) To provide care and support by strengthening programming for orphans and vulnerable children **infected**/**affected** by AIDS and by expanding life skills training for young people. |
| Ref. Trans. | | (c)以加强协助艾滋病孤儿和被艾滋病感染/影响脆弱儿童的方案,以及扩大助益年轻人的生活技能培训方式,提供照顾和支助。 |
| Org. Trans. | 21.07 | (c)通过加强对艾滋病感染/受害的孤儿和脆弱儿童的方案和扩大对年轻人的生活技能培训,提供照顾和支助。 |
| Adv. TransFool | | **[c]** To provide care and support by strengthening programming for orphans and vulnerable children **Disabled**/ **afflicted** by AIDS and by expanding life skill training for young people. |
| Trans. | 9.22 | [c]通过加强为孤儿和受艾滋病影响的弱势儿童提供照顾和支助,并扩大对年轻人的生活技能培训。 |
| Adv. WSLS | | (c) To provide **nursing** and **unstinted_**support by strengthening **i_Lifetv** for orphans and **susceptable** children infected/affected by **CPR_mannequins** and by **broadening** life skills training for young people. |
| Trans. | 12.57 | (c)通过加强孤儿和受CPR_迷彩感染/影响的易受感染儿童的i_Lifetv,并为年轻人提供更广泛的生活技能培训,提供护理和无毒的支助。 |
| Adv. kNN | | ( **so**) **address** provide care and support by strengthening **prioritization** for orphans and vulnerable children infected/affected by AIDS and by expanding life skills **issue** for young people. |
| Trans. | 5.63 | 因此,通过加强对艾滋病感染/受害的孤儿和脆弱儿童的优先事项和扩大对年轻人的生活技能的问题,解决提供照顾和支助。 |
| Adv. Seq2Sick | | (c) To provide care and support by strengthening **digital** for **dress** and **harmful** children **Journal**/ **Letter** by **Region** and by **disappear Violence** skills training for young people. |
| Trans. | 14.99 | (c)通过加强服装和有害儿童的数字,按区域分发新闻/信,并为年轻人提供暴力技能培训,提供照顾和支持。 |

## F    Effect of Back-Translation Model Choice on WSLS Performance

WSLS uses a back-translation model for crafting an adversarial example. In (Zhang et al., 2021), the authors investigate the En-De task and use the winner model of the WMT19 De-En sub-track (Ng et al., 2019) for the back-translation model. However, they do not evaluate their method for

Table 23: Performance of WSLS (En-De) with two back-translation models.

| Back-Translation | ASR | RDBLEU | RDchrF | Sim. | Perp. | #Queries |
|---|---|---|---|---|---|---|
| Marian NMT | 44.33 | 0.50 | 0.19 | 0.86 | 219.32 | 1262 |
| (Ng et al., 2019) | 51.68 | 0.58 | 0.21 | 0.81 | 241.96 | 1307 |

En-Fr and En-Zh tasks. To evaluate the performance of WSLS in Table 3, We have used pre-trained Marian NMT models for all three back-translation models. In order to show the effect of our choice of back-translation model, we compare the performance of WSLS for the En-De task when we use Marian NMT or (Ng et al., 2019) as the back-translation model in Table 23. As this Table shows, WSLS with Marian NMT as the back-translation model results in even more semantic similarity and lower perplexity score. On the other hand, WSLS with (Ng et al., 2019) as the back-translation model has a slightly more success rate. These results show that our choice of back-translation model does not highly affect the performance of WSLS.

## G    License Information and Details

In this Section, we provide some details about the datasets, codes, and models used in this paper. We should note that we used the models and datasets that are available in HuggingFace transformers (Wolf et al., 2020) and datasets (Lhoest et al., 2021) libraries.[9] They are licensed under Apache License 2.0. Moreover, we used PyTorch for all experiments (Paszke et al., 2019), which is released under the BSD license[10].

### G.1    Datasets

**WMT14**    In the Ninth Workshop on Statistical Machine Translation, WMT14 was introduced for four tasks. We used the En-De and En-Fr news translation tasks. There is no license available for this dataset.

**OPUS-100**    OPUS-100 is a multilingual translation corpus for 100 languages, which is randomly sampled from the OPUS collection (Tiedemann, 2012). There is no license available for this dataset.

### G.2    Models

**Marian NMT**    Marian is a Neural Machine Translation framework, which is mainly developed by the Microsoft Translator team, and it is released under MIT License[11]. This model uses a beam size of 4.

**mBART50**    mBART50 is a multilingual machine translation model of 50 languages, which has been introduced by Facebook. This model is published in the Fairseq library, which is released under MIT License[12]. This model uses a beam size of 5.

---

[9]These two libraries are available at this GitHub repository: https://github.com/huggingface.

[10]https://github.com/pytorch/pytorch/blob/master/LICENSE

[11]https://github.com/marian-nmt/marian/blob/master/LICENSE.md

[12]https://github.com/facebookresearch/fairseq/blob/main/LICENSE

### G.3 CODES

**kNN**    In order to compare our method with kNN (Michel et al., 2019), we used the code provided by the authors, which is released under the BSD 3-Clause "New" or "Revised" License.[13]

**Seq2Sick**    To compare our method with Seq2Sick (Cheng et al., 2020a), we used the code published by the authors.[14] There is no license available for their code.

**WSLS**    We implemented and evaluated WSLS (Zhang et al., 2021) using the source code published by the authors.[15] There is no license available for this GitHub repository.

## H    HUMAN EVALUATION

We conduct a preliminary human evaluation campaign of TransFool, kNN, and Seq2Sick attacks on Marian NMT (En-Fr) in the white-box setting. We randomly choose 90 sentences from the test set of the WMT14 (En-FR) dataset with the adversarial samples and their translations by the NMT model. We split 90 sentences into three different surveys to obtain a manageable size for each annotator. We recruited two annotators for each survey. For the English surveys, we ensure that the annotators are highly proficient English speakers. Similarly, for the French survey, we ensure that the annotators are highly proficient in French.

Before starting the rating task, we provided annotators with detailed guidelines similar to (Cer et al., 2017; Michel et al., 2019). The task is to rate the sentences for each criterion on a continuous scale (0-100) inspired by WMT18 practice (Ma et al., 2018) and Direct Assessment (Graham et al., 2013; 2017). For each sentence, we evaluate three aspects in three different surveys:

- *Fluency*: We show the three adversarial sentences and the original sentence on the same page (in random order). We ask the annotators how much they agree with the *"The sentence is fluent."* statement for each sentence.

- *Semantic preservation*: We show the original sentence on top and the three adversarial sentences afterwards (in random order). We ask the annotators how much they agree with the *"The sentence is similar to the reference text."* statement for each sentence.

- *Translation quality*: Inspired by monolingual direct assessment (Ma et al., 2018; Graham et al., 2013; 2017), we evaluate the translation quality by showing the reference translation on top and the translations of three adversarial sentences afterwards (in random order). We ask the annotators how much they agree with the *"The sentence is similar to the reference text."* statement for each translation.

We calculate 95% confidence intervals by using 15K bootstrap replications. The results are depicted in Figure 4. These results demonstrate that although the adversarial examples generated by Trans-Fool are more semantic-preserving and fluent than both baselines. According to the provided guide to the annotators for semantic similarity, the score of 67.8 shows that the two sentences are roughly equivalent, but some details may differ. Moreover, a fluency of 66.4 demonstrates that although the generated adversarial examples by TransFool are more fluent than the baselines, there is still room to improve the performance in this regard.

We follow the direct assessment strategy to measure the effectiveness of the adversarial attacks on translation quality. According to (Ma et al., 2018), since a sufficient level of agreement of translation quality is difficult to achieve with human evaluation, direct assessment simplifies the task to a simpler monolingual assessment instead of a bilingual task. The similarity of the translations of the adversarial sentences with the reference translation is shown in Figure 4c. The similarity of Seq2Sick is worse than other attacks. However, its similarity in the source language is worse. Therefore, we compute the decrease of similarity (between the original and adversarial sentences)

---

[13]The source code is available at `https://github.com/pmichel31415/translate/tree/paul/pytorch_translate/research/adversarial/experiments` and the license is avialable at `https://github.com/pmichel31415/translate/blob/paul/LICENSE`

[14]The source code is available at `https://github.com/cmhcbb/Seq2Sick`.

[15]https://github.com/JHL-HUST/AdvNMT-WSLS/tree/79945881f75d92ae44e9ebc10500d8590c09bb13

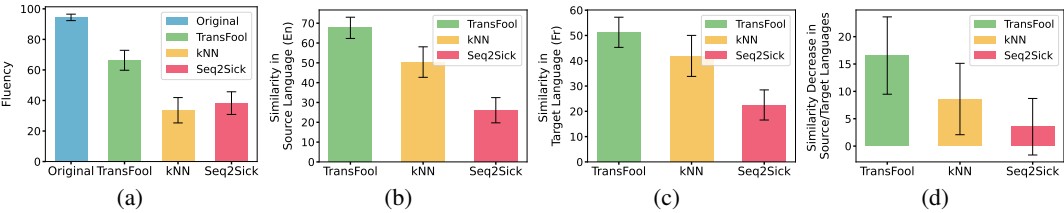

Figure 4: Human evaluation results for TransFool, kNN, and Seq2Sick attacks against Marian NMT (En-Fr).

from the source language to the target language. The results in Figure 4d show that all attacks affect the translation quality and the effect of TransFool is more pronounced than that of both baselines.

Finally, we calculate Inter-Annotator Agreement (IAA). There are two human judgments for each sentence. We average both scores to compute the final score for each sentence. To ensure that the two annotators agree, we only consider sentences where their two corresponding scores are less than 30. We compute IAA in terms of Pearson Correlation coefficient instead of the commonly used Cohen's K since scores are in a continuous scale. The results are presented in Table 24. Overall, we conclude that we achieve a reasonable inter-annotator agreement for all sentence types and evaluation metrics.

Table 24: Inter-annotator agreement for human evaluation.

| Sentence Type | Fluency | Similarity in En | Similairty in Fr |
|---|---|---|---|
| Original | 0.68 | - | - |
| TransFool | 0.85 | 0.82 | 0.79 |
| kNN | 0.91 | 0.82 | 0.86 |
| Seq2Sick | 0.89 | 0.88 | 0.83 |

