# OpenReview forum: "TransFool: An Adversarial Attack against Neural Machine Translation Models"
_ICLR.cc/2023/Conference — Submitted to ICLR 2023_

### Official Review · Reviewer_8Cx6 · 2022-10-24

**Confidence:** 5
**Correctness:** 3
**Technical Novelty And Significance:** 2
**Empirical Novelty And Significance:** 2
**Recommendation:** 3

**Clarity, Quality, Novelty And Reproducibility:**

Clarity: This paper is well-written and easy to follow.
Quality: The soundness of their method and evaluation is limited. Please refer to the weakness part.
Novelty: This paper is not very novel. Please refer to the weakness part.
Reproducibility: The reproducibility will not be a issue. The authors claims that their code will be released. And they provide the link of the metric and scripts used in their paper.


**Strength And Weaknesses:**

Pros:
+ This paper is good written and easy to follow.
+ The authors valid their method not only on research models but also on real world commercial product, like google translation.
+ The transferability analysis section is enlightening. It is not widely studied in NLP field. The transferability between different language is an important issue in multilingual NLP filed.


Cons:
- Novelty is limited.
a) First, the authors consider not only the attack success rate but also the fluency and semantic similarity. However, these two issues are widely studied in NLP attack field. For example, [1] has systematically reveal that the fluency and semantic similarity issue in textual adversarial attack and they empirically study the threshold of filtering the adversarial examples according to the fluency and semantic similarity.
b) Second, this paper claims that they propose a new strategy to incorporate the embedding vectors of a language model. However, utilizing language model to attack NLP model is not a novel technique. For example, [2-3] utilize BERT to conduct word-level substitution.

- Lack of soundness. This is not only the weakness of this paper but also the weakness of a large portion of textual adversarial attack works.
a) The soundness of their method is weak. This paper use language model due to it provide a meaningful representation of the tokens. However, based on the finding of previous work [1], the semantic similarity cannot be guaranteed with language model. A good example is that “I [like] eating apple.” and “I [hate] eating apple” have very high representation similarity according to the language model representation. But their semantic are totally opposite. So I am worried that the adversarial attack will change the semantic meaning of source sentence, leading to the over-estimation of attack success rate.
b) The soundness of their evaluation is weak. The authors evaluate the semantic similarity by the universal sentence encoder and BERTScore. However, as I mentioned above, the soundness of such model-based automatic metrics is limited. The high attack success rate could be partially due to the changing of meaning in source sentence. Conducting a human annotation experiment will be better than just showing an example.

[1] John Morris, Eli Lifland, Jack Lanchantin, Yangfeng Ji, Yanjun Qi ,Reevaluating Adversarial Examples in Natural Language, EMNLP 2020.
[2] Linyang Li, Ruotian Ma, Qipeng Guo, Xiangyang Xue, Xipeng Qiu, BERT-ATTACK: Adversarial Attack Against BERT Using BERT, EMNLP 2020.
[3] Siddhant Garg, Goutham Ramakrishnan, BAE: BERT-based Adversarial Examples for Text Classification, EMNLP 2020.


**Summary Of The Paper:**

This paper proposes a white-box adversarial attack against neural machine translation model named TransFool. TransFool uses an optimization loss function with three terms: a) adversarial loss to maximize the loss of the target NMT model; b) a similarity term to ensure that the adversarial example is similar to the original sentence; c) and loss of a language model to generate fluent and natural adversarial examples. The authors compare their TransFool method with two NMT attack methods, kNN and seq2sick, with respect to the Attack Success Rate, Relative decrease of translation quality, Semantic Similarity, Perplexity score and Token Error Rate. The proposed method can outperform baseline methods on two NMT model, Marian NMT and mBART50 MNMT model. The authors also show that the adversarial examples they found can conduct transfer attack on black box NMT models, using the adversarial examples from Marian NMT to attack mBART50 MNMT model and google translation api.

**Summary Of The Review:**

This paper proposes an adversarial attack method against neural machine translation model. The authors conduct experiments to show the superiority of their method. However, the novelty and the soundness of this paper is limited. So I tend to reject this paper.

---

> ### Author Response · Authors · 2022-11-16
> **Response to Reviewer 8Cx6 (Part 2/2)**
>
> > Lack of soundness. This is not only the weakness of this paper but also the weakness of a large portion of textual adversarial attack works.
>
> Due to the discrete nature of textual data, generating adversarial examples that are grammatically correct and can maintain semantic similarity is a difficult task. Also, evaluating the generated adversarial examples for these two criteria is not straightforward, as shown in the literature. Therefore, we agree with the reviewer that there may be some shortcomings in these regards in the field of adversarial attacks against textual data. In this paper, we reported many automatic metrics to evaluate TransFool compared to other baselines. In addition, we extended our study to span a broader range of the trade-offs between attack success rate and similarity/fluency thresholds. Moreover, we added human evaluation to the paper to provide more insights. We believe that more works are still necessary from the community to improve the quality of adversarial examples and evaluate them better. Yet, despite the current limitations in this research field, we should highlight that our attack outperforms state-of-the-art adversarial attacks in the literature in terms of many current automatic measures and in our human evaluation. Therefore, we believe that this work contributes to the field towards developing general solutions to design proper and efficient attacks against NMT models, and towards revealing further insights about their vulnerabilities.
>
> ***
> > This paper use language model due to it provide a meaningful representation of the tokens. However, based on the finding of previous work [1], the semantic similarity cannot be guaranteed with language model. A good example is that “I [like] eating apple.” and “I [hate] eating apple” have very high representation similarity according to the language model representation. But their semantic are totally opposite. So I am worried that the adversarial attack will change the semantic meaning of source sentence, leading to the over-estimation of attack success rate.
>
> We agree with the reviewer that by using the embeddings of a language model, we cannot completely guarantee semantic similarity. BERTScore (Zhang et al., 2019) conducted a comprehensive human evaluation. They showed that their proposed similarity metric based on the embeddings of a pretrained language model correlates highly with human evaluations. The automatic metrics and the human evaluation (added to the appendix of the revised manuscript) demonstrate that TransFool is able to maintain similarity to some degree, and it outperforms baselines in this regard. We also computed similarity in terms of BLEURT-20 [4], which is also shown to correlate better with human judgment than other automatic metrics [5], and the results for Marian NMT are as follows:
>
> | | En-Fr | En-De |En-Zh|
> |-|-|-|-|
> | TransFool | 0.65| 0.67| 0.67|
> | kNN | 0.61| 0.61| 0.66|
> | Seq2Sick|0.60| 0.52| 0.54|
>
> These show the same trend as the metrics reported in the paper. The results of this new metric for all cases are added to appendices D and E of the manuscript.
>
> Moreover, we should note that the similarity of the two sentences "I like eating apple." and "I hate eating apple." with a sentence encoder is 0.58, which is smaller than the generated adversarial examples (more than 0.8) by TransFool. Therefore, we believe that these automatic metrics still reveal the trend of semantic similarity to some degree.
>
> [4] Sellam, Thibault, et al. "Learning to evaluate translation beyond English: BLEURT submissions to the WMT metrics 2020 shared task." Proceedings of the Fifth Conference on Machine Translation.
>
> [5] Freitag, Markus, et al. "Results of the WMT21 metrics shared task: Evaluating metrics with expert-based human evaluations on TED and news domain." Proceedings of the Sixth Conference on Machine Translation. 2021.
>
> ***
> > The authors evaluate the semantic similarity by the universal sentence encoder and BERTScore. However, as I mentioned above, the soundness of such model-based automatic metrics is limited. The high attack success rate could be partially due to the changing of meaning in source sentence. Conducting a human annotation experiment will be better than just showing an example.
>
> We have been conducting human evaluation since the submission of the paper. The complete setup and results can be found in appendix H of the revised version of the manuscript. The summary of the results is as follows:
>
> |           | Fluency | Similarity |
> |-----------|---------|------------|
> | TransFool | 66.42   | 67.83      |
> | Knn       | 33.43   | 50.36      |
> | Seq2Sick  | 38.05   | 25.92      |
>
> These results show that compared to the baselines, the adversarial examples generated by TransFool are more fluent and more similar to the original sentence. The human evaluation confirms the results of automatic metrics in the view that our attack outperforms both baselines in terms of similarity and fluency.

---

> ### Author Response · Authors · 2022-11-16
> **Response to Reviewer 8Cx6 (Part 1/2)**
>
> We thank the reviewer for their valuable comments. We address each point raised by the reviewer hereunder. We hope that it will encourage discussion if any point remains unclear.
>
> > Novelty is limited
>
> We explained in detail in the general response the difference between this paper and other adversarial attacks in the literature of adversarial attacks against NLP systems in general. We kindly ask the reviewer to refer to the Novelty section of the general response. In summary, incorporating different constraints with gradient-based optimization is challenging since we have to compute gradients of the constraints w.r.t the target model embeddings. We proposed a new adversarial attack based on gradient-based optimization that can incorporate different differentiable constraint functions.
>
> To the best of our knowledge, there is *no similar* approach in the literature. We hope that our explanations have addressed the novelty concerns of the reviewer.
> ***
>
> > First, the authors consider not only the attack success rate but also the fluency and semantic similarity. However, these two issues are widely studied in NLP attack field. [1] has systematically reveal that the fluency and semantic similarity issue in textual adversarial attack and they empirically study the threshold of filtering the adversarial examples according to the fluency and semantic similarity.
>
> We thank the reviewer for bringing this work to our attention [1]. Indeed, there is a trade-off between semantic preservation/fluency and success rate. By considering only the adversarial examples with a similarity higher than a threshold, the success rate decreases as the threshold increases, and the quality of the adversarial examples increases. We added new experiments (appendix D of the revised manuscript) to study these effects and develop a more comprehensive evaluation, as it is indeed difficult to set a given threshold a priori in a general manner.
>
> If we evaluate the similarity by the sentence encoder mentioned in the paper [1], the success rate with different threshold values for similarity in the case of Marian (En-Fr) is as follows:
>
> |            | 0.80   | 0.85   | 0.90  | 0.95 |
> |------------|--------|--------|-------|------|
> | TransFool  | 42.94  | 32.92  | 22.04 | 9.62 |
> | kNN        | 16.53  | 11.82  | 6.81  | 1.54 |
> | Seq2Sick   | 3.34   | 1.94   | 1.10  | 0.23 |
>
> We did the same analysis for fluency. As suggested in paper [1], we count the grammatical errors by LanguageTool for the original sentences and the adversarial examples. We find the success rate for different thresholds  of the number of added grammatical errors by adversarial perturbations:
>
> |            | 0      | 1      | 2     |
> |------------|--------|--------|-------|
> | TransFool  | 29.98  | 53.96  | 64.81 |
> | kNN        | 8.65   | 22.47  | 32.65 |
> | Seq2Sick   | 10.28  | 20.37  | 25.44 |
>
> These interesting analyses show that with tighter constraints, we can generate better adversarial examples while the success rate decreases.
>
> All in all, according to the above new results, TransFool outperforms other works in the literature for different thresholds of similarity and grammatical errors. We hope that this extended analysis is helpful in providing a more comprehensive study that properly considers the trade-offs between attack success rates, and similarity/fluency metrics, which are hard to quantify exactly.
>
> ***
> > This paper claims that they propose a new strategy to incorporate the embedding vectors of a language model. However, utilizing language model to attack NLP model is not a novel technique. For example, [2-3] utilize BERT to conduct word-level substitution.
>
> Most of the papers in the literature of adversarial attacks against NLP systems, including the cited papers by the reviewer [2,3], use masked language models to find suitable replacements for modified words in the adversarial examples. As opposed to computer vision, these methods are not gradient-based, which may result in their suboptimal performance. However, we propose an optimization problem in the embedding space of the NMT model. In order to use the embeddings of a language model in the optimization problem in a differentiable manner, the language model and the NMT model need to have the same tokenizer. Moreover, we need to find a transformation between the embedding of the NMT and the language model. Due to these two reasons, we propose a new strategy to incorporate the language model with our attack. We train a language model on top of a fully-connected layer, which gets the embedding vectors of the NMT model in the input. This new method helps us to find a transformation between the two embedding spaces so that we can solve the proposed optimization problem and find the gradients w.r.t. the NMT embeddings. Both baselines in the white-box settings use the NMT embeddings for similarity due to these difficulties.

---

### Official Review · Reviewer_ZrSf · 2022-10-24

**Confidence:** 4
**Correctness:** 4
**Technical Novelty And Significance:** 3
**Empirical Novelty And Significance:** 3
**Recommendation:** 6

**Clarity, Quality, Novelty And Reproducibility:**

The writing is clear, and the approach is well-designed.

TransFool is novel as an adversarial attack algorithm for machine translation. The individual terms are not that new as there are prior works leveraging language models to improve the fluency of adversarial examples in the NLP domain, but this work provides a more comprehensive study for machine translation problems.

The authors plan to release their code for reproducibility.

**Strength And Weaknesses:**

Strengths:
1. Each term in the loss function of TransFool is well-motivated and properly designed.

2. The empirical study is pretty thorough. To me, the most interesting finding is that the generated adversarial examples can transfer to different target languages.

Weaknesses:

1. The study of defense is lacking. For example, it is helpful to see how the attack works with existing defenses against adversarial examples for language models. Also, it is interesting to try adversarial training with TransFool adversarial examples.

2. This is more of a question rather than a weakness, but have you tried TransFool for targeted attacks, and how does that work?

3. It is good to investigate more into the transferability between different target languages. For example, have you done these experiments on Google Translate?

**Summary Of The Paper:**

This work proposes TransFool for generating non-targeted adversarial examples against neural machine translation models. One core idea is to utilize an autoregressive language model (GPT-2) to add a language model loss term, which helps generate fluent adversarial examples. They also add a similarity loss, which constrains the distances of embedding distances between the original input sentences and adversarial examples. They perform a comprehensive study to evaluate their approach on translating from English to different target languages, and show that TransFool achieves higher attack success rates compared to baselines, while the generated adversarial examples better preserve the semantic meaning and look more natural. Meanwhile, they show that the generated adversarial examples transfer to other models in the black-box setting, including Google Translate, and can also transfer to different target languages.

**Summary Of The Review:**

This work proposes a well-designed adversarial attack for neural machine translation models, though it is not entirely novel. The paper presents a comprehensive study in different attack settings. In particular, the transferability of attacks between different target languages is interesting and new. Therefore, I lean towards accepting this work.

-------------
I thank the authors for explanation and adding more experiments, and I keep my original score.

---

> ### Author Response · Authors · 2022-11-16
> **Response to Reviewer ZrSf**
>
> We thank the reviewer for their valuable comments. We address each point raised by the reviewer hereunder:
>
> > The study of defense is lacking. For example, it is helpful to see how the attack works with existing defenses against adversarial examples for language models. Also, it is interesting to try adversarial training with TransFool adversarial examples.
>
> In this paper, our focus is to use TransFool to measure the robustness of NMT models and show that they are vulnerable to adversarial attacks. Since the proposed attack is transferable and stronger than previous methods, it reveals the susceptibility of NMT models to a deeper extent. We expect that our method could be used to measure and analyse the vulnerability of robust systems or defense methods. We however feel that this would be a different analysis. In that respect, and suggested by the reviewer, using TransFool to make NMT models more robust, e.g., with adversarial training, is very interesting, and we consider it as future research. However, we believe extending the adversarial attacks and defenses from language models/classification to NMT requires more thought and consider it as a future direction.
>
> ***
> > This is more of a question rather than a weakness, but have you tried TransFool for targeted attacks, and how does that work?
>
> We are extending the proposed attack for the targeted settings, in which the adversary aims to insert a target keyword into the translation (Cheng et. al. 2020). The preliminary results for inserting the target keyword "guerre" (war in French) into the translation is as follows:
>
> |           | ASR   | Sim  | Perp.  | TER   |
> |-----------|-------|------|--------|-------|
> | TransFool | 99.29 | 0.83 | 103.46 | 9.15  |
> | Seq2Sick  | 86.68 | 0.73 | 179.70 | 16.24 |
>
> These preliminary results seem to support the strength of our attack, and we will certainly conduct a more detailed evaluation in that respect in the future. Many thanks for the suggestion.
>
> ***
> > It is good to investigate more into the transferability between different target languages. For example, have you done these experiments on Google Translate?
>
> We thank the reviewer for this suggestion. We attacked Google Translate API for En-De translation by using the Marian (En-Fr) NMT mode as the reference model. The results are as follows:
>
> |           | ASR   | RDBLEU | RDchrF | SIM. | Perp.  | WER   |
> |-----------|-------|--------|--------|------|--------|-------|
> | TransFool | 67.42 | 0.65   | 0.26   | 0.85 | 198.56 | 20.78 |
>
> These results show that TransFool is highly transferable. We added this new experiment to Appendix E of the revised manuscript.
>
> ***
> > The individual terms are not that new as there are prior works leveraging language models to improve the fluency of adversarial examples in the NLP domain, but this work provides a more comprehensive study for machine translation problems. it is not entirely novel.
>
> We explained in detail in the general response the difference between this paper and other adversarial attacks in the literature of adversarial attacks against NLP systems in general. We kindly ask the reviewer to refer to the Novelty section of the general response. In summary, most of the works in the literature are not gradient-based, as opposed to computer vision, which may result in their subpar performance. Moreover, incorporating different constraints with gradient-based optimization is challenging since we have to compute gradients of the constraints w.r.t the target model embeddings. We proposed an adversarial attack based on gradient-based optimization that can incorporate different differentiable constraint functions.
>
> To the best of our knowledge, there is *no similar* approach in the literature. We hope that our explanations have addressed the concerns of the reviewer.

---

### Official Review · Reviewer_Jg82 · 2022-10-24

**Confidence:** 4
**Correctness:** 3
**Technical Novelty And Significance:** 3
**Empirical Novelty And Significance:** 2
**Recommendation:** 6

**Clarity, Quality, Novelty And Reproducibility:**

- New loss is proposed that consists of three terms. The ideas for these are out there for some time.
- New architecture is proposed to find adversarial sequences. It is an interesting modification of a common one.

I also imagine that the observed behaviour can be related to the requirement to include a projection layer in self-supervised learning or in other settings.

**Strength And Weaknesses:**

Strengths:
- Quality metrics are better than that of competitors.
- Experiments are interesting and numerous. They include human evaluation as well.
- Nice adversarial example found in Table 2

Weakness, methods:
- The novelty is limited. The paper contains some engineering tricks to make everything work, but all ideas are similar to those previously discussed in the literature.
- Not clear why we need the FC part in the approach. We can use BERT-like architectures to generate similar differentiable $v_i$-s as well. What would be the difference in this case?
- Investigation on the distortion of the generated adversarial sequence $x'$ is limited and includes only "Semantic Similarity" from Yang et al. Can you include metrics like BERT score for a pair of $x$ and $x'$ as well?


**Summary Of The Paper:**

The paper proposes a new approach for adversarial attacks on NMT models.
The authors design a way to propagate the gradient for embeddings and propose a specific loss that targets reasonable criteria.
The propagation is based on the common idea of relaxation with a new additional FC part to create embeddings for comparison.
The metrics are good, and overall, the results are promising.

**Summary Of The Review:**

The results seem promising, on the other side, the novelty is limited. From the paper, it is not clear, why this approach works in the NMT-attacks field, where other approaches failed.

---

> ### Author Response · Authors · 2022-11-16
> **Response to Reviewer Jg82**
>
> We thank the reviewer for their valuable comments. We address each point raised by the reviewer in this rebuttal:
>
> **Weaknesses:**
>
> > The novelty is limited. The paper contains some engineering tricks to make everything work, but all ideas are similar to those previously discussed in the literature.
>
> With all due respect, we tend to disagree with the reviewer and rather claim that our proposal of a new optimisation-based attack algorithm is not a mere trivial combination of engineering tricks. Even more, the proposed method achieves state-of-the-art results in adversarial attacks, which is not trivial either.
>
> We explained in detail in the general response the difference between this paper and other adversarial attacks in the literature of adversarial attacks against NLP systems in general. We kindly ask the reviewer to refer to the Novelty section of the general response. In summary, most of the works in the literature are not gradient-based, as opposed to computer vision, which may result in their subpar performance. Moreover, incorporating different constraints with gradient-based optimization is challenging since we have to compute gradients of the constraints w.r.t the target model embeddings. We proposed a novel adversarial attack based on gradient-based optimization that can incorporate different differentiable constraint functions.
>
> To the best of our knowledge, there is *no similar* approach in the literature. We hope that our explanations have addressed the novelty concerns of the reviewer.
>
> ***
> > Not clear why we need the FC part in the approach. We can use BERT-like architectures to generate similar differentiable $v_i$-s as well. What would be the difference in this case?
>
> The proposed optimization problem is in the embedding space of the NMT model. In order to use the embeddings of a language model in the optimization problem in a differentiable manner, the language model and the NMT model need to have the same tokenizer. Moreover, we need to find a transformation between the embedding of the NMT and the language model. Due to these two reasons, we propose to train a language model on top of a fully-connected layer, which gets the embedding vectors of the NMT model in the input. This method helps us to find a transformation between the two embedding spaces so that we can build a novel solution to the proposed optimization problem and find the gradients w.r.t the NMT embeddings. We should note that fixing the embedding layer of the language model with the NMT ones (instead of using the FC layer) results in low performance of the language model, and hence, low performance of the attack.
>
> Due to these difficulties, both white-box baselines, i.e., kNN and Seq2Sick,  define the similarity constraint in the embedding space of the target model, which may not necessarily capture the relationship between tokens.
>
> ***
> > Investigation on the distortion of the generated adversarial sequence $x'$ is limited and includes only "Semantic Similarity" from Yang et al. Can you include metrics like BERT score for a pair of $x$ and $x'$ as well?
>
> We have computed the similarity with BERTScore and presented the results in appendix  D  (white-box) and E (black-box). The results show that the similarity of the adversarial examples generated by TransFool is more than 0.95 in all cases, which is better than both baselines.
> ***
> > I also imagine that the observed behaviour can be related to the requirement to include a projection layer in self-supervised learning or in other settings.
>
> We agree with the reviewer that the proposed method, which learns the transformation between the NMT and language model embedding spaces, helps us to incorporate different constraints in the attack. This approach results in the high performance of TransFool.
>
> ***
> > From the paper, it is not clear, why this approach works in the NMT-attacks field, where other approaches failed.
>
> As discussed in the introduction, since we successfully integrate language model embeddings in our optimization problem, our adversarial examples are more inclined to preserve semantics. The effect of this component is studied in Appendix C. Moreover, our approach is based on optimization as opposed to most of the works in the literature of NLP adversarial attacks in general, which are based on word replacement, which results in much higher performance. We believe that both aspects are key components of our contribution, which permits us to push further the limits of robustness analysis in NMT systems, so that more robust systems can be designed in the future.

---

### Official Review · Reviewer_6HVS · 2022-10-25

**Confidence:** 4
**Correctness:** 3
**Technical Novelty And Significance:** 2
**Empirical Novelty And Significance:** 3
**Recommendation:** 5

**Clarity, Quality, Novelty And Reproducibility:**

The space after figures and tables are being squeezed too much.


**Strength And Weaknesses:**

Strength

- The proposed method is simple and straightforward.
- The experiment results show superior performance in attack success rate and significant decrease in translation quality.
- This work also demonstrates the capability and efficiency of blackbox attack.


weaknesses

- Existing adversarial attack on NMT aims at improving the robustness of these models. However, in this work, achieving a high attack success rate seems to be the goal. Therefore, I’m wondering if TransFool is as effective as baselines in improving robustness. Or what is the desired use case for this method?
- Translation models use beam search or similar mechanisms to generate high-quality output. Is the proposed attack still effective when using these mechanisms?
- Missing human validation. Although automatic metrics show significant decrease in translation quality, I’m not convinced that the algorithm triggers incorrect translation. Maybe the translation is correct but has a very low BLEU or chrF score (i.e., the attack method is attacking the automatic metrics instead of the NMT model, which could also be an interesting finding).


**Summary Of The Paper:**

This paper defines a new optimization objective function which combines the fluency, similarity and translation error to adversarially attack machine translation models. A gradient projection algorithm is applied to solve this optimization. Experiment results show the proposed method outperforms baselines. The transferability is also examined.


**Summary Of The Review:**

The novelty of the proposed method is somewhat incremental.

The experiment part is strong in terms of datasets and models. The automatic evaluation metrics also show strong performance.

I think adding a human evaluation would reveal lots of insights.

---

> ### Author Response · Authors · 2022-11-16
> **Response to Reviewer 6HVS**
>
> We would like to thank the reviewer for their valuable comments. We address each point raised by the reviewer hereunder:
>
> > Existing adversarial attack on NMT aims at improving the robustness of these models. However, in this work, achieving a high attack success rate seems to be the goal. Therefore, I’m wondering if TransFool is as effective as baselines in improving robustness. Or what is the desired use case for this method?
>
> In this paper, our focus is to use TransFool to study and measure the robustness of NMT models and show that they are quite vulnerable to adversarial attacks. Since the proposed attack is transferable and stronger than previous methods, it gets further in characterizing the vulnerability of NMT models. In future work, we indeed plan to use TransFool to improve the robustness of NMT models, e.g., with adversarial training, which is the state-of-the-art solution for adversarial robustness in other applications like computer vision.
> ***
>
> > Translation models use beam search or similar mechanisms to generate high-quality output. Is the proposed attack still effective when using these mechanisms?
>
> The target NMT models used in the experiments use beam search to generate translation. Marian NMT uses a beam size of 4, and mBART50 uses a beam size of 5. We clarified this point in the revised manuscript.
> Moreover, we attack Marian NMT (En-Fr) with different beam sizes to study the effect of this parameter on the attack performance. The results are as follows:
>
> |        | ASR   | RDBLEU | RDchrF | SIM. | Perp.  | TER   |
> |--------|-------|--------|--------|------|--------|-------|
> | beam=1 | 69.28 | 0.58   | 0.24   | 0.85 | 172.18 | 13.70 |
> | beam=2 | 70.38 | 0.57   | 0.23   | 0.85 | 178.07 | 13.81 |
> | beam=3 | 68.67 | 0.56   | 0.23   | 0.85 | 176.39 | 13.92 |
> | beam=4 | 69.48 | 0.56   | 0.23   | 0.85 | 177.20 | 13.91 |
> | beam=6 | 68.07 | 0.56   | 0.23   | 0.85 | 174.77 | 13.80 |
>
> As it can be seen, beam size does not affect the attack performance significantly. This may be due to the fact that we use the training loss for adversarial loss, and beam size is only used during inference.
> ***
>
> > Missing human validation. Although automatic metrics show significant decrease in translation quality, I’m not convinced that the algorithm triggers incorrect translation. Maybe the translation is correct but has a very low BLEU or chrF score (i.e., the attack method is attacking the automatic metrics instead of the NMT model, which could also be an interesting finding).
>
> The point raised by the reviewer is very interesting. We have been conducting human evaluation since the submission of the paper. The complete setup and results can be found in appendix H of the revised version of the manuscript. The summary of the results is as follows:
>
> ||Fluency|Similarity in source language (En)|Similarity in target language (Fr)|Similarity Decrease in Source/Target language|
> |-|-|-|-|-|
> | TransFool| 66.42| 67.83| 51.33| 16.50|
> | Knn| 33.43| 50.36| 41.84| 8.52|
> | Seq2Sick| 38.05| 25.92| 22.34| 3.58|
>
> These results show that, compared to the baselines, the adversarial examples generated by TransFool are more fluent and more similar to the original sentence. Moreover, the similarity of the translation of the adversarial sentences with the reference translations also demonstrates that, indeed, all attacks affected the translation quality. The difference between the similarity of the original and adversarial sentences in the source language and their translations in the target language can be an estimate of the effect of the attack on the translation. TransFool could decrease the similarity in the target language (French) with respect to the similarity in the source language to a larger extent than the baselines.
>
> > The novelty of the proposed method is somewhat incremental.
>
> We explained in detail in the general response the difference between this paper and other adversarial attacks in the literature of adversarial attacks against NLP systems in general. We kindly ask the reviewer to refer to the Novelty section of the general response. In summary, most of the works in the literature are not gradient-based, as opposed to computer vision, which may result in their subpar performance. Moreover, incorporating different constraints with gradient-based optimization is challenging since we have to compute gradients of the constraints w.r.t the target model embeddings. We proposed a novel adversarial attack based on gradient-based optimization that can incorporate different differentiable constraint functions.
>
> To the best of our knowledge, there is *no similar* approach in the literature. We hope that our explanations have successfully addressed the concerns of the reviewer.

---

> ### Comment · Reviewer_6HVS · 2022-12-14
> **Thank you for providing additional results.**
>
> I read the author responses. Thank you for clarifying the beam search and providing human evaluation results. However, the novelty is still incremental, making this paper very borderline. Therefore I keep my score unchanged.

---

### Author Response · Authors · 2022-11-16
**General Response to Reviewers (Part 2/2)**

In addition to the individual responses, we address the two common points mentioned in the reviews hereunder:
***
**P1: Novelty**

Due to the discrete nature of textual data, most works in the literature of adversarial attacks against NLP systems are based on token modification [1,2]. There exist some attacks based on derivative-free optimization, e.g., particle swarm or genetic algorithm [3,4]. On the other hand, there are only a few methods based on gradient-based optimization [5,6].

The token modification and derivative-free optimization-based methods incorporate linguistic constraints such as semantic similarity and fluency to ensure that the generated adversarial examples are not detectable. They mainly use a masked language model to find suitable replacements, or they use a language model perplexity to set a constraint on fluency. However, these methods are less effective  and their efficiency remains subpar compared to adversarial attacks in computer vision. This gap may be due to the fact that these methods are not gradient-based.

In the optimization-based method, however, it is challenging to incorporate these constraints in a differentiable manner. Both [5,6] use the embedding space of the target model to define the similarity constraint in their proposed optimization problem. This is a straightforward solution, but the embedding space of the target model may not necessarily capture the relationship between tokens. Also, adding a fluency term is not possible in this case since it requires a language model. And there are two challenges when incorporating a language model with an optimization method. The language model and the NMT model need to have the same tokenizer. Moreover, there needs to be a transformation between the embeddings of the NMT and that of the language model so that we can find the gradients w.r.t the target model embeddings. We exactly try to address this challenge, and we propose to train a language model on top of a fully-connected layer that gets the embedding vectors of the NMT model in the input. This novel method helps us to find a transformation between the two embedding spaces so that we can solve the proposed optimization problem and find the gradients w.r.t. the NMT embeddings.

Overall, we propose an adversarial attack based on gradient-based optimization that can incorporate different differentiable constraint functions. The superiority of our attack evaluated by numerous automatic metrics, as well as human evaluation, confirms that the proposed approach takes similarity preservation and fluency constraints into account. Moreover, the optimization-based method instead of token modification results in a higher success rate.

To the best of our knowledge, there is no similar approach, which incorporates linguistic constraints in a differentiable manner, in the literature. We hope that the above explanations have addressed the reviewers' concerns regarding the paper's novelty.

[1] Morris, John, et al. "TextAttack: A Framework for Adversarial Attacks, Data Augmentation, and Adversarial Training in NLP." EMNLP 2020.

[2] Roth, Tom, et al. "Token-modification adversarial attacks for natural language processing: A survey." arXiv preprint arXiv:2103.00676 (2021).

[3] Alzantot, Moustafa, et al. "Generating Natural Language Adversarial Examples." EMNLP 2018.

[4] Zang, Yuan, et al. "Word-level Textual Adversarial Attacking as Combinatorial Optimization." ACL 2020.

[5] Sato, Motoki, et al. "Interpretable adversarial perturbation in input embedding space for text." IJCAI 2018.

[6] Cheng, Minhao, et al. "Seq2Sick: Evaluating the Robustness of Sequence-to-Sequence Models with Adversarial Examples." AAAI 2020.

***
**P2: Human Evaluation**

We have been conducting a preliminary human evaluation study since the submission of the paper. The complete setup and more elaborate results (i.e., confidence intervals and inter-annotator agreements) can be found in appendix H of the revised version of the manuscript. The summary of the results is as follows:

||Fluency|Similarity in source language (En)|Similarity in target language (Fr)|Similarity Decrease in Source/Target language|
|-|-|-|-|-|
| TransFool| 66.42| 67.83| 51.33| 16.50|
| Knn| 33.43| 50.36| 41.84| 8.52|
| Seq2Sick| 38.05| 25.92| 22.34| 3.58|

These results show that, compared to the baselines, the adversarial examples generated by TransFool are more fluent and more similar to the original sentence. The difference between the similarity of the original and adversarial sentences in the source language and their translations in the target language can be an estimate of the effect of the attack on the translation. TransFool could decrease the similarity in the target language (French) with respect to the similarity in the source language to a greater extent compared to the baselines.

---

### Author Response · Authors · 2022-11-16
**General Response to Reviewers (Part 1/2)**

We thank all reviewers for their comments. We are glad to receive positive feedback from reviewers, particularly:
- [R1, R2, R3 and R4]: The experiment part is strong and pretty thorough.
- [R1, R2 and R3]: The optimization objective function is new, well-motivated, and properly designed.
- [R1 and R2]: The automatic evaluation metrics show superior performance.
- [R3 and R4]: Transferability between different languages is interesting and enlightening.
- [R1 and R4]: The work demonstrates efficiency in black-box attack and transferability analysis.
- [R3 and R4]: The paper is well-written and easy to follow.
- [R3 and R4]: The method is validated on a real-world commercial product, like Google Translate.
- [R2]: The sample adversarial sentences are nice.

In addition to the above comments, we received valuable feedback from the reviewers, which helped us improve the quality of the paper. We implemented several new experiments according to the comments and addressed every point raised by reviewers in the individual responses. The new experiments for this rebuttal are as follows:
- [R1 and R4] Conducting human evaluation (Appendix H)
- [R1] Investigating the effect of beam search on the attack performance
- [R3] Evaluating transferability to different languages when attacking Google Translate (Appendix E)
- [R3] Preliminary experiments on the targeted scenario
- [R4] Adding a new automatic similarity metric (Appendices D and E)
- [R4] Trade-off analysis between success rate and similarity/grammar check (Appendix D)

---

### Decision · Program_Chairs · 2023-01-20

**Decision:**

Reject

**Justification For Why Not Higher Score:**

Although the author tried hard to address the issue, there are some inherent issues of the research in NLP adversarial attacks. In particular, the adversarial examples do not really keep the meaning and grammatical correctness of the source input.  Therefore, it's hard to argue that translation quality degradation is caused by adversarial attacks or simply because the translation model cannot deal with grammatically incorrect input.

Another issue is although the idea of the paper is interesting, it does not elaborate enough on the insight behind the proposed approach. Therefore, several reviewers concern the novelty of the paper.

**Justification For Why Not Lower Score:**

N/A

**Metareview: Summary, Strengths And Weaknesses:**

The paper presents a method for an adversarial attack machine translation model. In particular, the algorithms seek for small pertubation that can significantly degrade the translation quality. Multiple evaluation approaches, including human evaluation, are provided in the revision during the rebuttal. Overall, the paper is interesting and the authors tried hard to address all the comments. However, it's still arguable if the paper is in publishable status to a top venue.


Strengths:
+ Different from most existing papers that focus only on text classification tasks, the paper proposes an adversarial attack algorithm for machine translation. The community should encourage more papers considering adversarial attacks for more complex NLP tasks.
+ The paper shows significant improvement on the previous baseline, seq2sick, for attacking the machine translation model.

Weaknesses:

- Several reviewers pointed out the concern of novelty. Although the authors argue the novelty of the approach in the rebuttal, it is still unclear what is the key insight of the proposed approach and how it advances the field. I would suggest the paper elaborating on the key challenge of adversarial attack for translation and how the proposed methods address the challenge. Specifically, the description Sec. 4 provides a nice explanation what has been done, but does not articulate enough what are the intuition and insight behind the approach.

- A few reviewers point out the issue of similarity metrics, and the authors provide human evaluation in the rebuttal. Although I (and reviewers) appreciate the authors provide new results in such a short time, there are still some remaining issues. 1) The evaluation is done in a small scale, it is unclear if the results are reliable; it is also unclear how the annotators are selected and how they get paid. 2) The human evaluation results show that the approach does significant change the meaning of the source sentence. Although the authors argue that the propose approach has smaller difference between the similarity of the original and adversarial sentences in the source v.s target, it is not clear to me this is a right metric. The similarity might not be linearly scale and when the source sentence is grammatically incorrect, the translation can off the track.

- Related to the previous point, the Appendix D.4 shows adversarial examples generated by the proposed approaches and baselines. Although it demonstrates the proposed appraoch is better than baseline, none of adverarisl examples generated by Transfool make inceptible changes that keep the meaning and maintain the grammatical correctness. For example, Table 9,  the state is changed and there is a spacing issue. Table 10, 12 the meaning of sentences change significantly and the sentences are not grammatically correct. Table 11, 13, 14 the sentences are not grammatically correct. Table 15, 16, the changes are less perceptible, but still the meaning of sentences change. Table 15, the tense changed. Table 16 is probably the only example makes sense. Besides, those examples also show that the evaluation metric is not reliable. For example, the translation of Table 13 is not faithful to the source sentence. Table 16, the BLUE score of Transfool is low because it transliterates the name.

I agree with the authors that many of prior works suffer from similar issue. However, this does not justify that these concerns should be tolerated. Instead, the community should be more rigiorious in the definition and the evaluation of the advesarial examples.

**Summary Of Ac-Reviewer Meeting:**

N/A